# Exosomal secreted SCIMP regulates communication between macrophages and neutrophils in pneumonia

Xiaolei Pei [1,2,5] ✉, Li Liu[1,2,5], Jieru Wang[1,2,5], Changyuan Guo[3], Qingqing Li[3], Jia Li[1,2], Qian Ren[1,2], Runzhi Ma [1,2], Yi Zheng[3], Yan Zhang[3], Li Liu[4], Danfeng Zheng[3], Pingzhang Wang[3], Ping Jiang[4], Xiaoming Feng [1,2], Erlie Jiang [1,2], Ying Wang [3] ✉ & Sizhou Feng [1,2] ✉

In pneumonia, the deficient or delayed pathogen clearance can lead to pathogen proliferation and subsequent overactive immune responses, inducing acute lung injury (ALI). While screening human genome coding genes using our peripheral blood cell chemotactic platform, we unexpectedly find SLP adaptor and CSK interacting membrane protein (SCIMP), a protein with neutrophil chemotactic activity secreted during ALI. However, the specific role of SCIMP in ALI remains unclear. In this study, we investigate the secretion of SCIMP in exosomes (SCIMP^exo) by macrophages after bacterial stimulation, both in vitro and in vivo. We observe a significant increase in the levels of SCIMP^exo in bronchoalveolar lavage fluid and serum of pneumonia patients. We also find that bronchial perfusion with SCIMP^exo or SCIMP N-terminal peptides increases the survival rate of the ALI model. This occurs due to the chemoattraction and activation of peripheral neutrophils dependent on formyl peptide receptor 1/2 (FPR1/2). Conversely, exosome suppressors and FPR1/2 antagonists decrease the survival rate in the lethal ALI model. *Scimp*-deficient and *Fpr1/2*-deficient mice also have lower survival rates and shorter survival times than wild-type mice. However, bronchial perfusion of SCIMP rescues *Scimp*-deficient mice but not *Fpr1/2*-deficient mice. Collectively, our findings suggest that the macrophage-SCIMP-FPRs-neutrophil axis plays a vital role in the innate immune process underlying ALI.

SLP adaptor and CSK interacting membrane protein (SCIMP) was previously reported as a transmembrane adaptor protein that plays a role in major histocompatibility complex II (MHCII) signaling[1]. Additionally, it has been discovered to be a universal adaptor of Toll-like receptors (TLRs) in macrophages, although it is not essential for their function[2]. Furthermore, recent research has shown that SCIMP acts as a spatiotemporal transmembrane scaffold for Erk1/2 signaling in TLR-activated macrophages[3]. Surprisingly, while screening candidate genes for human encoding proteins using our peripheral blood cell chemotactic platform, we unexpectedly discovered that cell culture

[1]State Key Laboratory of Experimental Hematology, National Clinical Research Center for Blood Diseases, Haihe Laboratory of Cell Ecosystem, Hematopoietic Stem Cell Transplantation Center, Institute of Hematology & Blood Diseases Hospital, Chinese Academy of Medical Sciences & Peking Union Medical College, Tianjin 300020, P. R. China. [2]Tianjin Institutes of Health Science, Tianjin 301600, P. R. China. [3]Department of Immunology, School of Basic Medical Sciences and NHC Key Laboratory of Medical Immunology, Peking University, Beijing 100191, P. R. China. [4]Tianjin First Central Hospital, Tianjin Medical University, Tianjin 300192, P. R. China. [5]These authors contributed equally: Xiaolei Pei, Li Liu, Jieru Wang. ✉e-mail: peixiaolei@ihcams.ac.cn; yw@bjmu.edu.cn; szfeng@ihcams.ac.cn

supernatants from HEK293 cells overexpressing SCIMP could chemoattract peripheral neutrophils.

In pneumonia induced by the infection of bacteria, viruses, fungi or mycoplasma etc., the absence of pathogen invading signals released by alveolar macrophages or epithelial cells to neutrophils or monocytes in the peripheral usually leads to the pathogen proliferation and expansion, which is one of the main reasons for acute lung injury (ALI) or acute respiratory distress syndrome (ARDS)[4]. Unfortunately, at the early stage of pneumonia, non-pathogen sourced invading signals released by these residential cells are poorly identified.

The first stage of pneumonia progression is the invasion of pathogens, followed by the activation of alveolar epithelial cells (AEs) and alveolar macrophages (AMs) act as the frontline defense, which release signals and activate downstream immune cells, such as neutrophils[5,6], circulating monocytes/macrophages[4,7], dendritic cells[8,9] and adoptive immune cells[4,7,10]. If the sensitivity of the AEs/AMs to pathogens is reduced, the inner response and the related signals released by AEs/AMs would be delayed or absent, and consequently, the downstream immune cells could not arrive at the pulmonary tissue in a timely manner, allowing the titer or count of pathogens to increase.

Circulating neutrophils are the first type of immune cells recruited by chemoattractants after the invasion of pathogens. Rapid neutrophil chemotaxis results in the timely phagocytosis of pathogens[11]. Neutrophil chemotaxis is crucial to prevent the exponential growth of pathogens at the early stage of pneumonia. Delayed neutrophil chemotaxis can lead to pathogen proliferation and a vigorous downstream immune response[7,12,13]. However, the specific communication between alveolar resident macrophages and neutrophils during bacterial infection remains incompletely clear.

In this study, we found that SCIMP could be secreted in an exosomal manner and was elevated both in the bronchoalveolar lavage fluid (BALF) and serum of pneumonia patients compared with the control cohort, as well as in the lung tissue and BLAF of ALI model mice. Additionally, we found that macrophages were a source of exosomal SCIMP, and SCIMP protein could chemoattract peripheral neutrophils. Furthermore, we demonstrated that exosomal SCIMP mediates communication between resident macrophages and peripheral neutrophils during pneumonia. To fully understand the effect of SCIMP on pneumonia, we utilized chemotaxis assays, *Scimp*-deficient mice, *Fpr1/2*-deficient mice, and a lethal ALI murine model to elucidate the role of the macrophage-SCIMP-FPR-neutrophil axis in pneumonia.

## Results

### SCIMP protein has chemotactic activity and can be secreted via exosomes after bacterial stimulation

In our previous study[14], we found that cell culture supernatants from the SCIMP-overexpressing HEK293 cells could chemoattract peripheral neutrophils (Fig. 1a). As expected, the SCIMP protein could be detected in these supernatants with a C-terminal tag antibody, an anti-6×his antibody (Fig. 1b). Furthermore, the SCIMP protein-containing supernatants were separated into exosome precipitation and exosome-free supernatant by the ultracentrifuge separation method, and by western blot, SCIMP was mainly observed in the exosome component as well as the exosome biomarkers, CD9, CD63, CD81[15], and TSG101[16] (Fig. 1c, the details of the preparation and evaluation of SCIMP antibody are in the method section). SCIMP-positive exosomes (SCIMP^exo) were further verified by the nanoparticle tracking flow cytometer (NTF), after staining the exosomes [50–150 nm] with a PE-conjugated anti-CD63 antibody and FITC-conjugated anti-SCIMP antibody (Fig. 1d), as well as by transmission electron microscope (TEM) after staining the exosomes with a gold-particle conjugated anti-SCIMP antibody (Fig. 1e).

In the macrophage cell line (RAW264.7) under the stable condition, endogenous SCIMP protein could hardly be observed in the

exosomes from the supernatant, while endogenous SCIMP protein was detected in the exosomes from the RAW264.7 cells after stimulation with denatured *E. coli* (ATCC#19183, Multiplicity of infection, MOI = 100) at 2 and 6 h but not observed with Lipopolysaccharides (LPS) (Fig. 1f). Additionally, the endogenous SCIMP protein expression was significantly downregulated in the exosomes from the RAW264.7 cells with *Scimp* stably knocked down after stimulation with denatured *E. coli* at 2 and 6 h (Supplementary Fig. S1). In the exosomes purified from the RAW264.7 cells after stimulated with denatured *E. coli* (ATCC#19138, MOI = 100) for 2 h, the percentage of SCIMP-positive exosome in the total exosomes (CD63 positive) was significantly higher than that from the cell with PBS (Fig. 1g). Moreover, a murine pneumonia model was established by bronchial perfusion of *E. coli* ($2 \times 10^6$ colony-forming unit or CFU per mouse), and upregulated endogenous SCIMP protein expression was observed in the pulmonary tissue by immunohistochemistry (IHC, Fig. 1h).

To explore the mechanism underlying SCIMP secretion by exosome, the RAW264.7 cell line stably expressing the SCIMP-GFP fusion protein was established. Compared with the control cells, RAW264.7 cells stably expressing GFP, SCIMP-GFP protein was observed to colocalize with the exosome regulation protein, EEA1[17]/LAMP2A[18], at 0 h (Fig. 1i and Supplementary Fig. S2). while at 1 and 4 h after incubated with heat-denatured *E. coli* (MOI = 1000) in the RAW264.7 cell line stably expressing the SCIMP-GFP fusion protein, the colocalizations decreased (Fig. 1i and Supplementary Fig. S2), which might be due to the SCIMP protein was secreted outside the cells by exosome secreting pathway. Interestingly, no decrease of colocalizations of SCIMP-GFP and EEA1/LAMP2A was observed at the same time points after stimulated with LPS (Supplementary Fig. S2).

### The elevated level of SCIMP^exo in the serum and BALF of pneumonia patients indicates a correlation between SCIMP and pneumonia

To explore SCIMP-related diseases, we established the SCIMP^exo detection kit, which was composed of the beads conjugated with CD9 capture antibody and PE-anti-CD63 and FITC-anti-SCIMP antibodies (Fig. 2a, b, the details of the preparation and evaluation of the detection kit are in the methods section). BALF samples were collected from 36 pneumonia patients and 20 pulmonary tumor patients (Supplementary Table S1). Exosome extraction was carried out following a previously reported method[19] and SCIMP expression was assessed using the SCIMP^exo detection kit. By analyzing following the gating strategy (Fig. 2b), the mean level of SCIMP^exo (mean fluorescence intensity, MFI) in the pathogen-induced pneumonia cohort (Infected group) was significantly higher than that in the pulmonary tumor cohort (Uninfected group, Fig. 2c), and the area under the curve (AUC) of the receiver operating characteristic (ROC) curve was 0.83 (Fig. 2d). Considering the heterogeneity of BALF exosomes from the different individuals, the percentage of SCIMP^+ particles in CD63^+ particles was analyzed; this percentage was significantly higher in the pneumonia cohort than in the pulmonary tumor cohort (Fig. 2e), and the AUC of the ROC curve was 0.84 (Fig. 2f). According to the different types of infectious pathogens, the pneumonia cohort was divided into the bacteria subgroup, fungal subgroup, virus subgroup and mycoplasma subgroup. The SCIMP protein level in all these pathogen-related subgroups was universally higher than that in the pulmonary tumor cohort (Fig. 2g), similar results were observed for the percentage of SCIMP^+ particles among the CD63^+ particles in the BALF (Supplementary Fig. S3).

Serum samples were collected from 57 pulmonary-infected patients and 50 patients without pneumonia (Supplementary Table S2). After extracting the total exosomes from serum, the SCIMP protein level was measured by the SCIMP^exo detection kit. The mean level of SCIMP in the serum from the pathogen-induced pneumonia patients was also higher than that of the control subjects (Fig. 2h), with an AUC of 0.78. In the pneumonia cohort, the SCIMP level in the higher

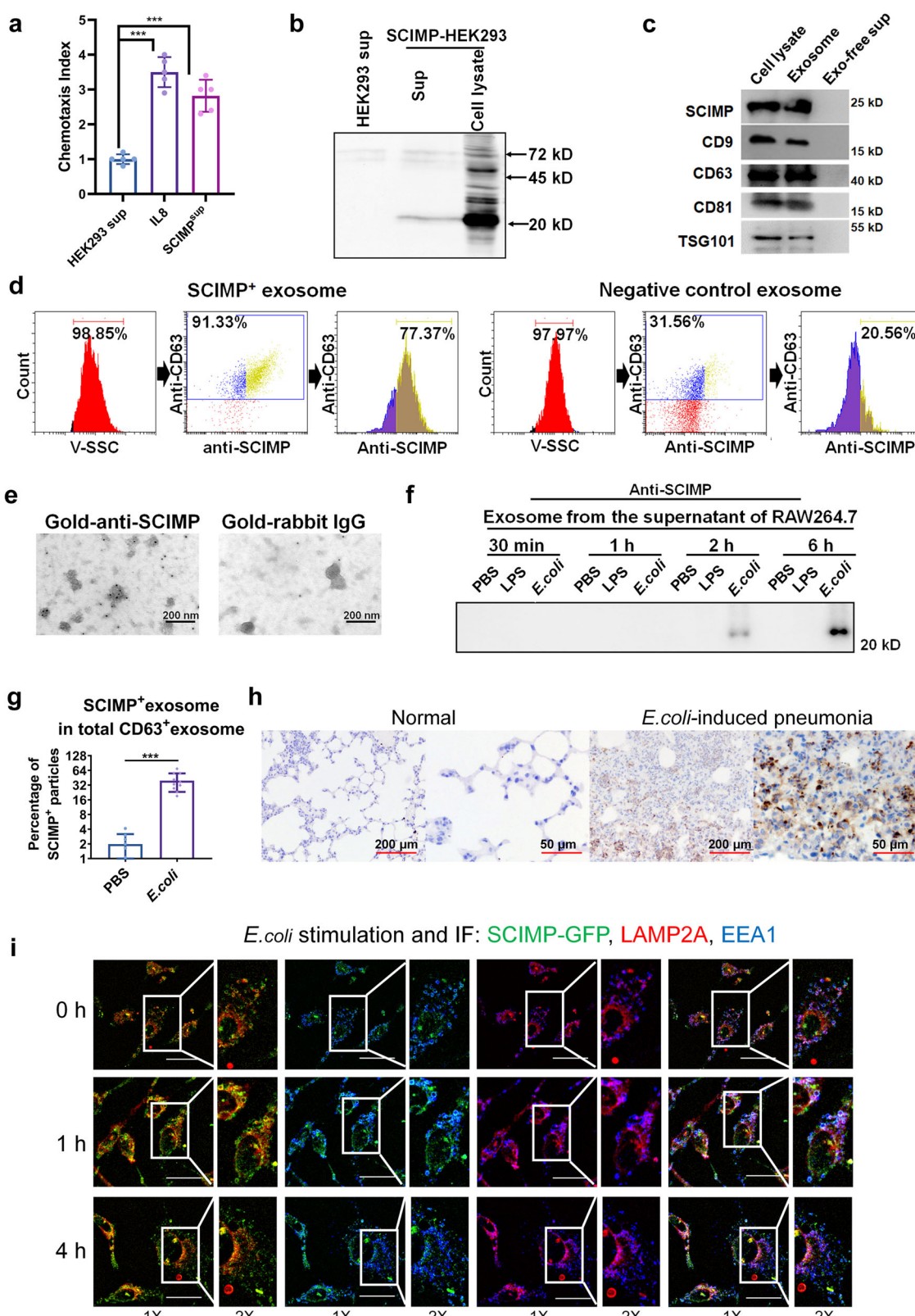

neutrophil subgroup (neutrophil percentage >70%) was higher than that in the lower neutrophil subgroup (neutrophil percentage <70%) (Fig. 2i). Moreover, the percentage of neutrophils in the higher SCIMP level subgroup (MFI > 2000) of the pneumonia cohort was higher than that in the lower SCIMP level subgroup (MFI < 2000, Fig. 2j). Compared with that in patients without pneumonia, the exosomal SCIMP protein level in pneumonia patients was elevated at approximately 2

weeks before fever and decreased 4 weeks after fever (Fig. 2l), which suggested that SCIMP might participate in the inflammatory response to pneumonia. Moreover, there were more white blood cells in the higher SCIMP level subgroup than in the lower SCIMP level subgroup (Supplementary Fig. S4), but similar results were not observed for monocytes (Supplementary Fig. S4), indicating that the correlation between SCIMP and the peripheral monocytes was low.

**Fig. 1 | SCIMP can be secreted in exosomes by macrophages stimulated with bacteria. a** The chemotactic activity of the SCIMP-containing supernatant ($n = 5$) from SCIMP-overexpressing HEK293 cells (SCIMP-HEK293) toward peripheral neutrophils was measured using a chemotaxis assay, with Interleukin 8 (IL-8, 10 ng/mL) used as the positive control ($n = 5$). **b** The exogenous SCIMP-6×his recombinant protein in the HEK293 cells could be detected by the anti-6×his antibody both in the cell culture supernatant and cell lysate via western blot. **c** The exosomes present in the cell culture supernatant of SCIMP-HEK293 were purified using the ultracentrifugation method, and the presence of SCIMP-6×his recombinant protein was confirmed by western blot analysis, along with the exosomal markers CD9, CD63, CD81 and TSG101. **d** The exosomes from SCIMP-HEK293 or the control HEK293 were analyzed by NTF, and the diameter of the particles was indicated by the V-SSC channel, and PE-anti-CD63 antibody and FITC-anti-SCIMP antibody were used as detection antibodies. **e** The purified exosomes from SCIMP-HEK293 were observed under the TEM, stained with gold-conjugated anti-SCIMP antibody (scale bar = 200 nm). **f** The endogenous SCIMP protein secreted in exosomes by RAW264.7 cells with the stimulation of PBS, LPS (1 μg/mL), or heat-denatured *E. coli* (MOI = 100) for 0.5, 1, 2, and 6 h, was detected by western blot using anti-SCIMP antibody. **g** The percentage of SCIMP-positive particles in the total exosomes (CD63 positive), which were purified from the supernatant of RAW264.7 cells treated with PBS ($n = 8$) or *E. coli* (MOI = 100, $n = 10$) for 2 h, were stained and measured by NTF. **h** The endogenous SCIMP protein in the pulmonary tissue of the mice with *E. coli*-induced pneumonia was detected by IHC using anti-SCIMP antibody (scale bar = 200 μm/50 μm). **i** The colocalization of SCIMP (exogenously expressed SCIMP-GFP recombinant protein, green), EEA1 (stained with anti-EEA1 antibody, blue), and LAMP2A (stained with anti-LAMP2A antibody, red) was observed by confocal microscopy (IF, Immunofluorescence) in RAW264.7 cells stimulated with heat-denatured *E. coli* (MOI = 1000) for 0, 1, and 4 h (scale bar = 20 μm). The raw data is available in the "Source Data".

## Exosome-related protein and SCIMP protein levels are elevated in the BALF of the *E. coli*-induced ALI model

To verify the observation of upregulated SCIMP expression in pneumonia patients, we established *E. coli*-induced ALI murine model and LPS-induced ALI murine model. Then, at 24 h, the exosome from the BALF were extracted by the size exclusion chromatography (SEC) method and analyzed by quantitative mass spectrometry (MS, Fig. 3a). As shown in Fig. 3b, the exosomal proteomic profile of the BALF from the *E. coli*-induced ALI model was partially different and partially overlapped with that from LPS-induced ALI model. Compared with the PBS-treated control mice, the functions of the highly expressed proteins in *E. coli*-induced ALI model mice were mainly enriched in neutrophils and antigen presentation analyzed by Gene Ontology (GO), while those in LPS-induced ALI model mice were mainly enriched in necrosis and pyroptosis (Fig. 3c). The enriched cellular components of the proteins that were expressed higher in the *E. coli*-induced ALI than that in the LPS-induced ALI model mainly included exosome production (Fig. 3d). The CD9, CD63, CD81 and SCIMP protein levels were significantly elevated in the *E. coli*-induced ALI model but not in the LPS-induced ALI model (Fig. 3e), which suggested a difference in the underlying cellular and molecular mechanisms between these two models. Furthermore, SCIMP^exo began to be secreted into the BALF at approximately 15 min in the *E. coli*-induced ALI model (Fig. 3f), which was earlier than the secretion of classical chemokines (Supplementary Fig. S5).

## SCIMP^exo have chemotactic activity and it depends on the extramembrane fragment

In previous studies by our team, several novel chemokine-like proteins, including PSMP[14,20–22], FAM19A5 and FAM3D[23–25] as well as their chemotaxis receptors, were identified. All these findings were based on a chemokine screening system[26]. The supernatant of SCIMPHEK293 cells containing abundant SCIMP^exo was also found to have the ability to chemoattract peripheral neutrophils (PBNs) and a weaker ability to chemoattract peripheral monocytes (PBMs) (Fig. 4a) with IL-8 and Stromal cell-derived factor 1 (SDF-1) as the positive control, but it could not chemoattract peripheral lymphocytes (PBLs) (Fig. 4a). By N-terminal sequencing of the SCIMP protein purified from the supernatant of SCIMP-HEK293, we found that the SCIMP protein contains 18 amino acids before the transmembrane domain at the N-terminus, which was the main alternative translation (Supplementary Fig. S6). By using the anti-6×his antibody to stain the SCIMP-6×his-overexpressing HEK293 cells with or without cell membrane penetration, we found 6×his located at the C-terminus of SCIMP was mainly located inside of the cell membrane (Supplementary Fig. S7). By using the FITC-conjugated SCIMP antibody and FITC-conjugated anti-6×his antibody in the SCIMP^exo detection kit, we found that the C-terminus of SCIMP was mainly at the inside of the exosomes purified from the supernatant of SCIMP-6×his-overexpressing HEK293 cells (Supplementary Fig. S8). Furthermore, the N-terminus truncated mutant (the first 17 amino acids of the N-terminus were absent) SCIMP^exo that lacked its extra-membrane domain lost its ability to chemoattract neutrophils in the TaxiScan assay (Fig. 4b, c), which means that the chemotaxis ability of SCIMP^exo relied on the extramembrane domain. Purified SCIMP N-terminus peptides (SCIMP^N,18aa) and Purified human SCIMP protein (SCIMP^pro) expressed and purified from the SCIMP-overexpressing BL21DE3 (the purity evaluation available in the supplementary material) showed similar abilities to SCIMP^exo in terms of chemoattracting neutrophils in the TaxiScan assay (Fig. 4d).

To test the in situ chemotaxis ability of SCIMP in vivo, we perfused the exosomes purified from a murine cell line (CHO cells), the exosomes carrying high murine SCIMP levels purified from murine SCIMP-overexpressing CHO cells (SCIMP-CHO), and the chemically synthesized and purified murine SCIMP N-terminus peptides (SCIMP^N, "MSWWRDNF") into the bronchi of wild-type C57 mice. Bronchial perfused (b.p) with exogenous murine SCIMP^exo or murine SCIMP^N effectively chemoattracted neutrophils into BALF (Fig. 4e, f). Additionally, murine SCIMP^exo and murine SCIMP^N significantly enhanced bacteria clearance (Fig. 4g, h) in vivo as well as the Reactive Oxygen Species (ROS) released from the primary neutrophils in vitro (Fig. 4i).

## SCIMP is sufficient and necessary to increase the survival rate of the *E. coli*-induced ALI model

Given the finding that SCIMP could be secreted in exosomes, as shown in Fig. 5a, the exosome inhibitor, GW4869, was used to inhibit exosome secretion in the lungs of ALI model. This inhibition led to more serious lung injury (Fig. 5b), decreased numbers of SCIMP⁺ exosomes (Fig. 5c) and significantly reduced survival rates (Fig. 5d), which were consistent with a previous report[27]. To explore the role that SCIMP plays in pneumonia progression, purified murine SCIMP^exo, purified murine SCIMP^pro (the details of purification and evaluation are available in the supplementary material) and murine SCIMP^N were perfused into the bronchia along with a lethal dose of *E. coli* (Fig. 5a). The lung injury in the groups treated with SCIMP components was significantly ameliorated (Fig. 5e), and the survival rate was significantly elevated (from 40% to 80%) in the *E. coli*-induced ALI model (Fig. 5f). At 4 h after the perfusion, the percentage of neutrophils in the BALF from the groups treated with murine SCIMP components was significantly higher than that in BLAF from the control groups (Fig. 5g, h), while the cell count of AMs and Infiltrating Macrophages (IMs) in BALF were not significantly changed (Supplementary Fig. S9). At 4 h after perfusion, the residual *E. coli* clone counts in the lungs were analyzed (Fig. 5i) and the groups treated with SCIMP components had much lower *E. coli* clone counts than the control groups (Fig. 5j).

## Secreted SCIMP is necessary for neutrophils recruitment to the lung in the *E. coli*-induced ALI model

Next, *Scimp*-deficient mice (*Scimp*^-/-) were used to establish the ALI model and immune cell infiltration, lung injury and survival were

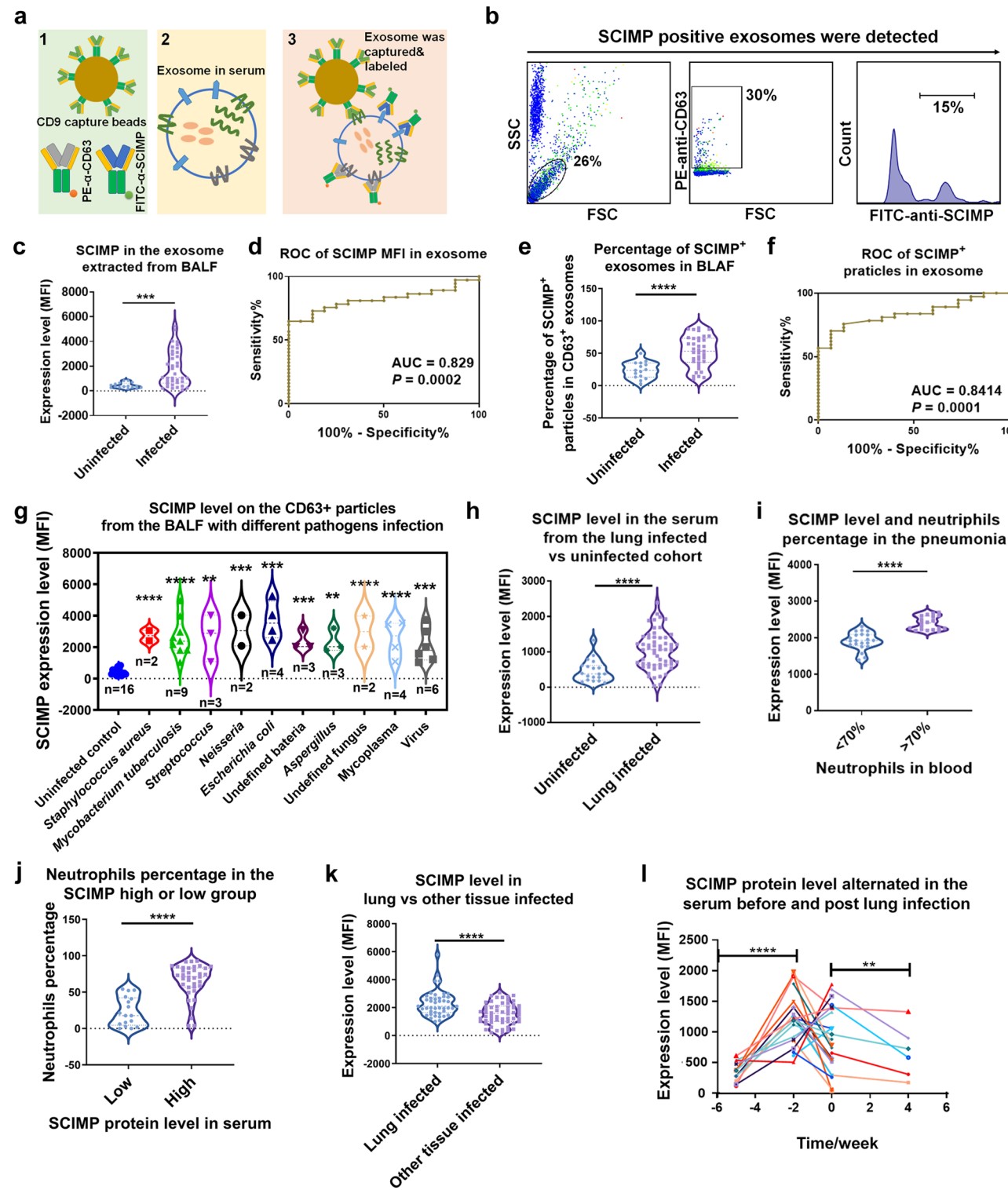

observed (Fig. 6a). Under the normal condition, the immune system of *Scimp^-/-* mice was not obviously different from wild-type mice. However, in the ALI model, the neutrophil count in the BALF was much lower than those in the wild-type mice at 2 and 4 h after bacteria perfusion, while the alveolar macrophage count did not change (Fig. 6b, c). At 4 h after perfusion, the bacterial clone count in the *Scimp^-/-* group was greater than that in the wild-type group, and the higher bacterial count in the *Scimp^-/-* group was rescued with the murine Scimp^N perfusion (Fig. 6d, e). In the *E. coli*-induced ALI model,

*Scimp^-/-* mice had exacerbated lung injury (Fig. 6f) and a reduced survival rate (Fig. 6g), which could be rescued by the perfusion of murine SCIMP^N.

## SCIMP^exo could chemoattract neutrophils and the chemotaxis activity is dependent on its N-terminus and formyl peptide receptors (FPRs)

To identify the receptors involved in SCIMP-mediated chemotaxis, the chemotaxis receptors in GPCR superfamily A were transiently

**Fig. 2 | The exosomal SCIMP protein in the pneumonia-related patients was detected using the SCIMP$^{exo}$ detection kit. a** The SCIMP detection kit is schematically shown. **b** The gating strategy for detecting SCIMP-positive exosomes is shown. **c**–**f** The relative expression level (**c**) and the percentage of SCIMP$^+$ particles in CD63$^+$ particles (**e**) in BALF samples from patients with pulmonary infection ($n = 36$) or without pulmonary infection ($n = 20$) were measured using the SCIMP$^{exo}$ detection kit; The ROC curves were generated based on the exosomal SCIMP expression level (**d**, $p$ = false positive rate) or SCIMP$^+$ particles percentage (**f**, $p$ = false positive rate) in the pulmonary-infected ($n = 36$) and -uninfected groups ($n = 20$), and the AUC was calculated. **g** The expression level of exosomal SCIMP in CD63$^+$ exosome in BALF samples from pulmonary-infected patients was statistically compared according to the pathogen types. **h** The exosomal SCIMP expression level of the serum samples from the pulmonary-infected ($n = 57$)/uninfected ($n = 50$) patients were measured and statistically compared. **i** The serum samples from the pulmonary-infected patients were subgrouped based on the neutrophil percentage in the white blood cells (>70%, $n = 30$ or <70%, $n = 27$), and the exosomal SCIMP expression level of the serum samples in these two subgroups was measured and analyzed. **j** The serum samples from the pulmonary-infected patients were subgrouped according to the exosomal SCIMP expression level in the serum (>2000, $n = 40$ or <2000, $n = 17$) and the neutrophils percentage in the white blood cells in these two subgroups was measured and analyzed. **k** The exosomal SCIMP expression level of the serum samples from the patients with pulmonary infection ($n = 57$) and those with other tissue infections ($n = 40$) was measured and compared. **l** The exosomal SCIMP expression level of serum samples collected at different time points from pulmonary-infected patients ($n = 34$) were measured, with time 0 being when the patient was first diagnosed with infectious fever. The raw data is available in the "Source Data".

overexpressed in HEK293 cells as previously reported[26], and fortunately, FPR1/2-overexpressing HEK293 cells (FPR1/2-HEK293) but not the empty plasmid transfected HEK293 cells could be chemoattracted by the purified full-length SCIMP protein (The details of purification and evaluation are available in the supplementary material) in a dose-dependent manner (Fig. 7a) but not be chemoattracted by the control protein that expressed and purified with the same method to SCIMP protein (Supplementary Fig. S10). And incubating the cells with N-Formyl-Met-Leu-Phe (fMLF, one agonist of FPRs) could desensitize the chemoattractant activity of the SCIMP protein to FPR1/2-HEK293 cells with a dose-dependent manner (Fig. 7b). FPR1/2 exogenously expressed on the HEK293 cell membrane could be internalized when incubated with the SCIMP protein (Fig. 7c, d). By the stimulation with SCIMP protein, the calcium flux in the cytoplasm of FPR1/2-HEK293 cells but not the empty plasmid transfected HEK293 cells could be activated and continuing with fMLF stimulation, no calcium flux was observed in FPR1/2-HEK293 cells, indicating desensitization of the cells by the treatment of SCIMP protein (Fig. 7e, f and Supplementary Fig. S11). The ligand-receptor binding assay, including the saturation binding assay and competitive binding assay, were performed. The affinity constant of SCIMP$^N$ binding to FPRs (Kd) was 3.8 nM (FPR1)/ 3.5 nM (FPR2), and the Kd of murine SCIMP$^N$ binding to murine FPR1/2 was 1.08 nM (murine FPR1)/0.54 nM (murine FPR2) (Fig. 7g and Supplementary Fig. S12). The IC50 values of human or murine SCIMP$^N$ and fMLF competitive binding to the FPRs was 55.2 nM (FPR1)/38.4 nM (FPR2)/148.2 nM (murine FPR1)/43.1 nM (murine FPR2) (Fig. 7g and Supplementary Fig. S12).

### The FPRs antagonist can inhibit the SCIMP-mediated chemotaxis to neutrophils in vivo

To confirm that the chemotaxis activity of SCIMP on neutrophils in vivo occurred in the FPRs-dependent manner, the *C57* mice were injected with the FPRs antagonist (Cyclosporin H, CsH), and 1 h later the exogenous SCIMP$^{exo}$, murine SCIMP$^N$ and exosomes purified from the supernatant of bone marrow-derived macrophages (BMDM) from the WT or *Scimp$^{-/-}$* mice were bronchial perfused, and 4 h later the counts of total neutrophils (CD11b$^+$Ly6G$^+$F4/80$^-$) and FPRs$^+$ neutrophils (CD11b$^+$Ly6G$^+$F4/80$^-$FPRs$^+$) in the BALF were measured by cytometry (Fig. 8a). By following the gating strategy shown in Supplementary Fig. S13, the results showed that after FPRs antagonist pretreatment, the chemotaxis of exogenous murine SCIMP$^{exo}$ (Fig. 8b, e) and murine SCIMP$^N$ (Fig. 8c, f) to neutrophils or FPRs$^+$ neutrophils was significantly inhibited, while the percentage of infiltrating classical macrophages (IMs, CD11b$^+$CD11c$^-$F4/80$^+$) that migrated from circulation or resident alveolar macrophages (AMs, CD11b$^+$CD11c$^+$F4/80$^+$) was not altered by the SCIMP$^N$ (Supplementary Fig. S14). Moreover, the exosomes from *Scimp$^{-/-}$* BMDMs showed a reduced chemotaxis activity to total neutrophils and FPRs$^+$ neutrophils, compared with the exosomes from WT BMDMs (Fig. 8d, g).

### FPRs are necessary for the effect of SCIMP on the ALI model and neutrophils in vivo

The *Fpr1/2* had been reported to be mainly expressed on the mature neutrophils in the periphery[28]. In this study, we established the *E. coli*-induced ALI model on WT or *Fpr1/2*-deficient mice (*Fpr1/2$^{-/-}$*), and rescued ALI by perfusing murine SCIMP$^N$ via bronchus, and observed neutrophil recruitment, bacterial clearance, pulmonary inflammation and the survival of ALI model (Fig. 9a). Compared with WT mice, the chemotaxis activity of murine SCIMP$^N$ to neutrophil was abolished in the *Fpr1/2$^{-/-}$* mice (Fig. 9b, c). The bacterial clone count in the BALF from the *Fpr1/2$^{-/-}$* ALI model was much higher than that in the BALF from the WT ALI model mice, and perfusing with murine SCIMP$^N$ could not enhanced bacterial clearance (Fig. 9d, e). Furthermore, compared with the *Fpr1/2$^{-/-}$* ALI model perfused with PBS, *Fpr1/2$^{-/-}$* ALI model mice perfused with SCIMP$^N$ exhibited similar lung injury and survival rate (Fig. 9f, g). These data suggested that *Fpr1/2* is essential for the function of SCIMP in rescuing lethal ALI model.

### Discussion

The migration or infiltration of immune cells, including neutrophils and macrophages, which are vital for the immune response to pathogen invasion, has been proven to be related to inflammation induced local elevated exosome levels[29–31]. However, the direct chemotactic activity of the membrane proteins on exosomes has not been previously reported. Here, we first showed that SCIMP is secreted in exosomes. We demonstrated that the secreted SCIMP is carried by exosomes but not present in the exosome-free cell culture supernatant by western blotting, NTA and TEM assays, mainly depending on the SCIMP antibody. And we produced and purified the SCIMP antibody by following the designed processes to guarantee the specificity of SCIMP antibody to recognize the SCIMP protein in the bacteria lysate, cell lysate, and exosome. The observation that the exosome secretion inhibitor, GW4869, reduced SCIMP secretion into BALF supports the finding the SCIMP secreting manner. Additionally, exosomal SCIMP could also be detected in serum and BALF from patients with lung infection. In this study, we purified the full-length human SCIMP expressed by bacteria and HEK293T cells. Even though the location of these differently sourced SCIMP proteins on SDS-PAGE was discrepant (the location of SCIMP protein produced by bacteria was near 15 KD, while the location of SCIMP protein produced by SCIMP-overexpressing HEK293T cells or secreted by murine macrophage cell line was near 20 KD), both demonstrated chemotaxis activity to neutrophils. This might due to the modification of the overexpressed protein in the eukaryotic cells.

Peripheral blood neutrophils were found to be the main chemotactic target cells of SCIMP both in vitro and in vivo, and SCIMP could also chemoattract monocytes but not lymphocytes. Moreover, we found the extracellular domain of SCIMP (and N-terminus peptides) played an essential role in chemotaxis, which resulted from the loss of

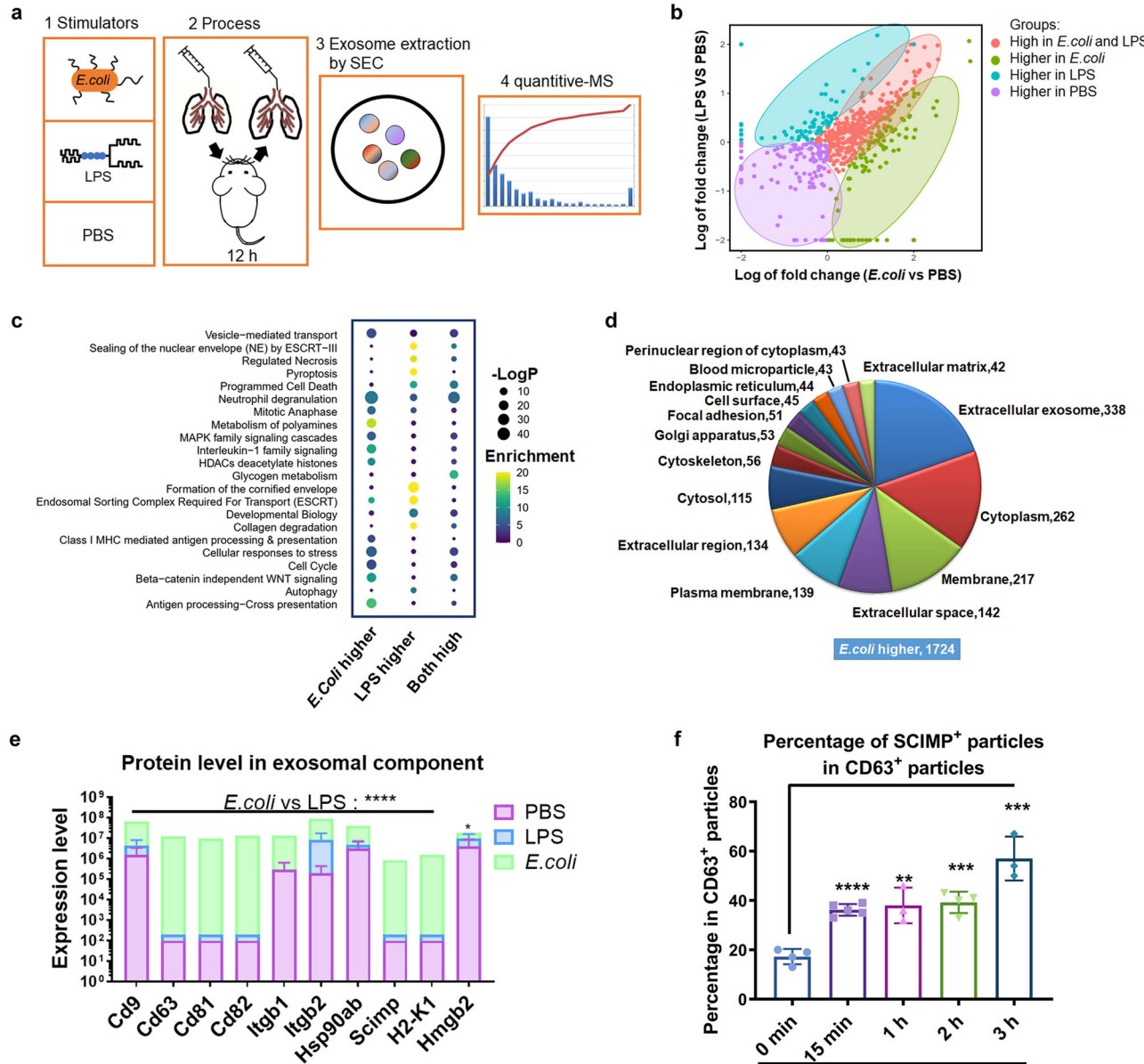

**Fig. 3 | In the ALI models, *E. coli* induced more exosomes to be released into the BALF than LPS. a** The schematic workflow for proteomic analysis of the exosomes extracted from the BALF in the ALI models using different stimulators. **b** According to the MS results, the proteins distribution in the two-dimensional plot was based on the fold change of the LPS group versus PBS group and the *E. coli* group versus PBS group; The proteins were classified into four subgroups including high protein level in the *E. coli*-treated subgroup (green), high protein level in the LPS-treated subgroup (blue), high protein levels in the PBS-treated subgroup (purple), and high protein levels in both the *E. coli*-treated subgroup and LPS-treated subgroup (red).

**c** The enriched functions were analyzed (P is estimating significance) and shown based on the alteration of protein level in three different treatment groups. **d** The proportion of the subcellular localizations enriched by the proteins in the subgroup with higher expression in the *E. coli*-treated mice that those in LPS-treated mice was calculated. **e** The level of exosomal proteins, including SCIMP, in the BALF of the three groups were summarized (*n* = 3). **f** In the *E. coli*-induced ALI model, BALF was collected at 0 min, 15 min, 1 h, 2 h, and 3 h after *E. coli* perfusion (*n* = 4), and the percentage of SCIMP-positive particles was measured using the SCIMP[exo] detection kit. The raw data is available in the "Source Data".

chemotactic ability to neutrophils when the extracellular domain of exosomal SCIMP was truncated. The results of the ligand and receptor interaction assays supported the finding that FPR1/2 are the chemotactic receptors of SCIMP N-terminus peptides.

In the ALI murine model, we demonstrated that SCIMP[N] could increase the survival rate, accelerate CD11b[+]Ly6G[+] neutrophils infiltration, and help with bacterial clearance, but they did not influence the percentage of CD11b[+]CD11c[+]F4/80[+] AM cells in the lung. The oppositive phenotype of ALI model was observed in the *Scimp*-deficient mice as well as *Fpr1/2*-deficient mice. The administration of SCIMP[N] improved the survival rate of the *Scimp*-deficient mice but not

*Fpr1/2*-deficient ALI model mice. Although the effect of neutrophils on ALI in this study was positive, it was also necessary to explore the negative impact of the uncontrolled number and overreaction of neutrophils on inflammation and ALI survival.

According to the published reports, exosomes can transport signals or mediate multiple physical and pathological processes, including tumor cell survival[32], tumor metagenesis[33], immune response[34] and tissue differentiation[35]. The membrane proteins found on exosomal membrane might contribute to exosome secretion, exosome structure, exosome targeting and exosome adhesion[36]. The materials that are encapsulated by exosomes have been screened by multiple high-

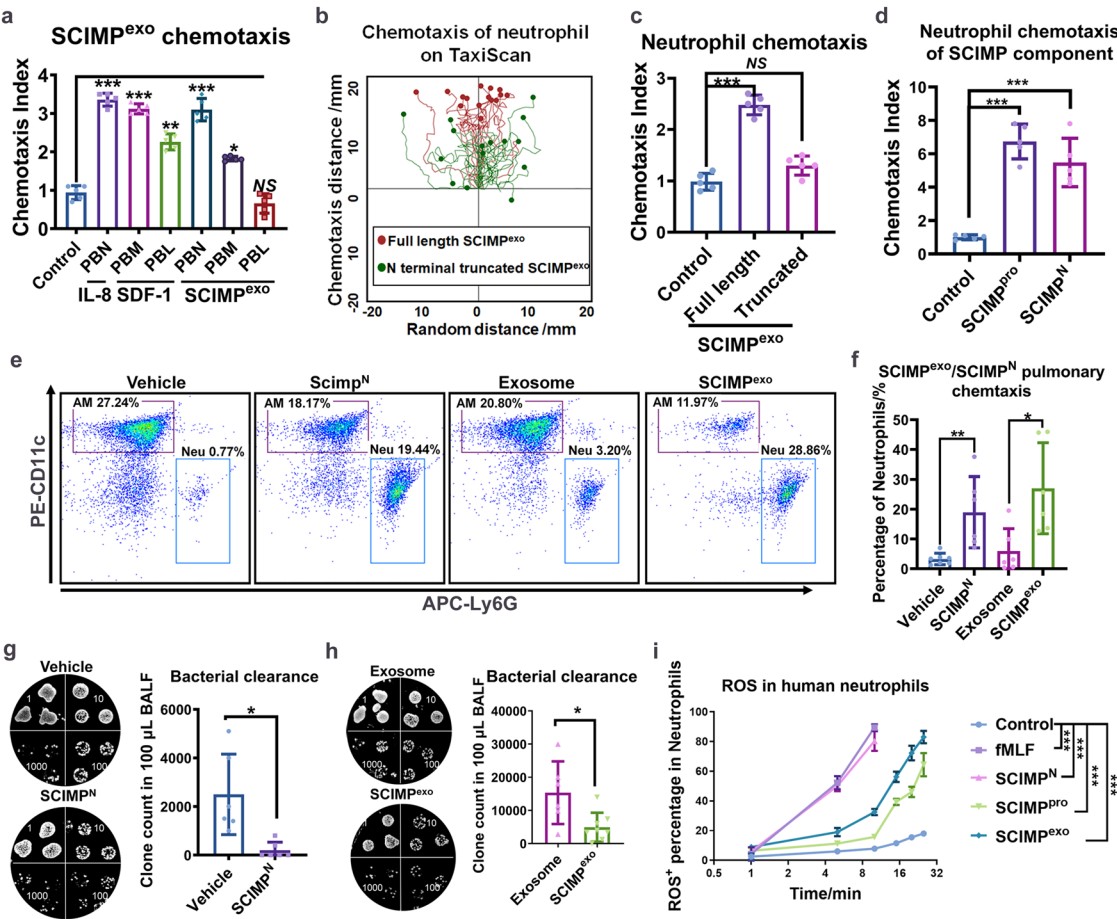

**Fig. 4 | SCIMP can chemoattract the peripheral neutrophils both in vitro and in vivo. a** After separating human peripheral blood neutrophils (PBN), monocytes (PBM), and lymphocytes (PBL), the chemotactic activity of 1 nM IL-8, 1 nM SDF-1, and 1 mg/mL SCIMP-positive exosomes (SCIMP$^{exo}$) to these cells in vitro were measured using the TaxiScan system ($n = 5$). **b, c** The chemotaxis ability of the full-length SCIMP$^+$ exosomes (1 mg/mL) and the extramembrane N-terminus truncated SCIMP$^+$ exosomes (1 mg/mL) to human peripheral blood neutrophils in vitro were measured using the TaxiScan system ($n = 5$); SCIMP$^-$ exosomes (1 mg/mL) acted as the control in (**c**). **d** The chemotactic activity of the purified full-length SCIMP protein (1 nM) and the SCIMP$^N$ (1 nM) to human peripheral blood neutrophils in vitro were measured using the TaxiScan system ($n = 5$). **e, f** The pulmonary in situ chemotaxis activity of the murine SCIMP$^N$ (10 μg in 50 μL PBS per mouse), vehicle buffer (PBS, 50 μL), murine SCIMP$^{exo}$ (10 mg in 50 μL PBS per mouse), and the

control exosomes, purified from CHO cells (10 mg in 50 μL PBS per mouse), was determined by measuring the percentage of neutrophils (Neu) in the BALF at 4 h post bronchial perfusion of these stimulators ($n = 6$). **g, h** After the mice were pretreated the mice with murine SCIMP$^N$ (10 μg in 50 μL PBS per mouse), SCIMP-negative exosomes (10 mg in 50 μL PBS per mouse), or SCIMP-positive exosomes (10 mg in 50 μL PBS per mouse) via b.p for 2 h, the bacteria (*E. coli*) count in the BALF was measured at 4 h after the bacteria bronchial perfusion ($1 \times 10^6$ CFU per mouse, $n = 6$ in each group). **i** The activity of fMLF (1 nM), SCIMP$^N$ (1 nM), SCIMP protein (1 nM), and SCIMP$^{exo}$ (1 mg/mL) on the ROS release of human peripheral blood neutrophils was detected using cytometry at the time points of 1, 5, 10, 15, 20, 25 min after the stimulation ($n = 2$). The raw data is available in the "Source Data".

throughput methods, such as mass spectrometry[37], the next-generation RNA sequencing[38].

Previous reports revealed that SCIMP acts as a nonspecific TLRs adaptor and could transduce signals from TLRs to downstream molecules[2]. Interestingly, we found that the SCIMP protein in macrophages could be secreted and served as a novel innate immune response mediator between resident alveolar macrophages and circulating neutrophils under the conditions of bacterial invasion in the lung. Although TLRs on resident macrophages are the well-known sensors of both bacteria and LPS, the colocalization of SCIMP with EEA1/LAMP2A was observed in the stable state, while in RAW264.7 cells incubated with bacteria, the colocalization of SCIMP with EEA1/LAMP2A decreased, possibly due to the exosomal secretion of SCIMP, but this phenomenon was not observed in the cells incubated with LPS. In the subsequent experiments, SCIMP$^N$ elevated the survival rate of *E. coli*-induced ALI model but not the LPS-induced ALI model (data not shown). This finding indicates that different stress signals are induced in macrophages by bacteria and LPS. Another process is

inflammasome secretion, which can be independently induced after pathogen stimulation but not LPS stimulation[39].

In our previous study[26], we found a series of proteins had predicted transmembrane domains in their amino acid sequence, could be secreted and was able to chemoattract neutrophils and monocytes. To date, exosomes, as functional units, have not yet been reported to have chemoattracting activity but have been reported to act as inflammatory units. In the in vitro chemotaxis assay, the SCIMP$^{exo}$ secreted into the supernatant might generate a concentration gradient similar to classical chemokine concentration gradients. Therefore, in this study, the exosome-specific roles of SCIMP were described using a series of experiment to illuminate the ligand-receptor relationship between SCIMP and FPRs; this relationship may be the mechanism by which SCIMP-positive exosomes interact with FPRs-positive cells. According to previous reports, FPRs, including FPR1 and FPR2, are highly expressed by neutrophils and macrophages; in human body, the expression level of FPR1 was much higher than FPR2, while in mouse the expression level of

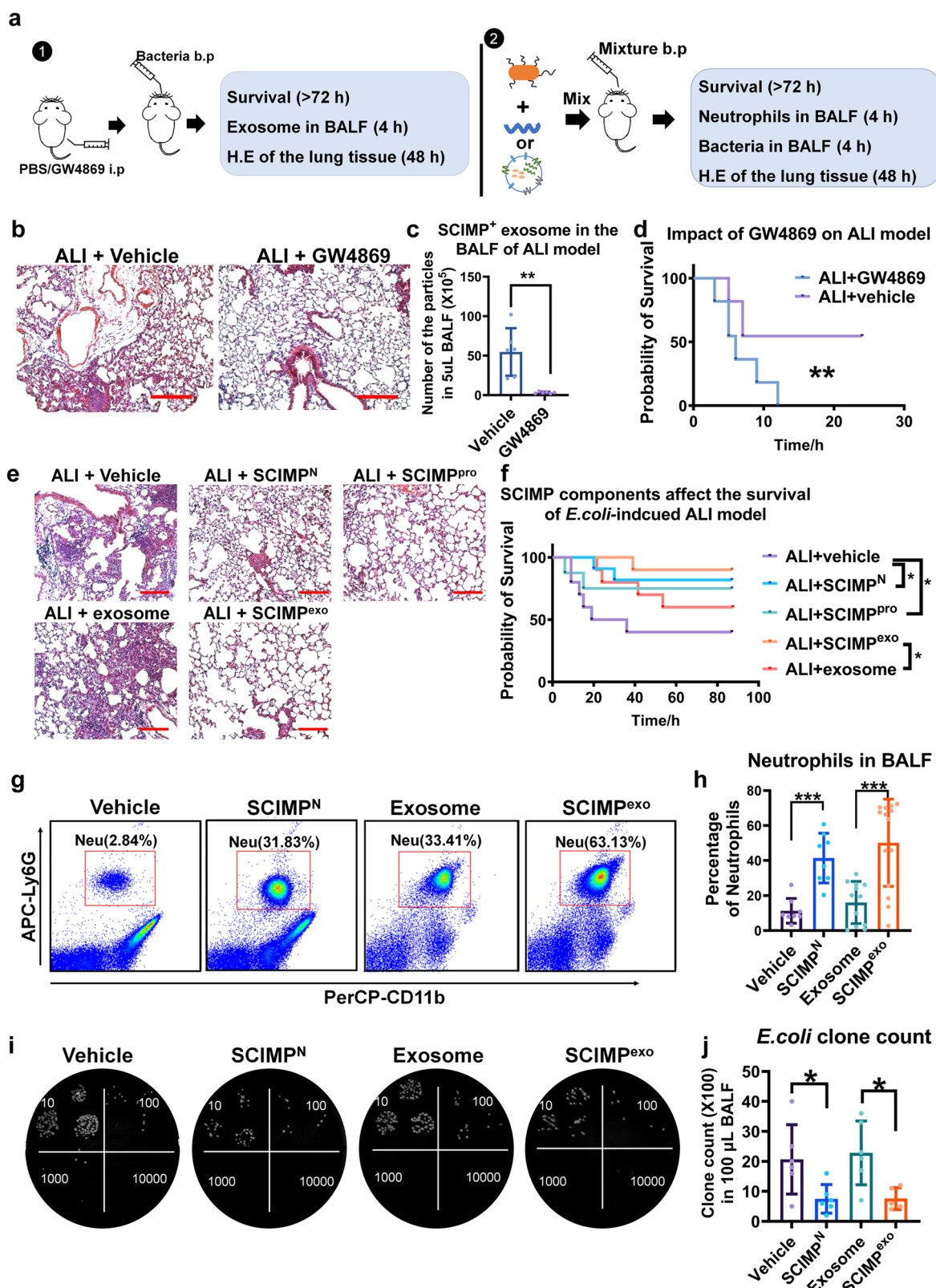

murine FPR2 was much higher than murine FPR1[40]. FPRs were found to be highly related to multiple types of inflammation and were reported to be a receptor of plague on host immune cells[41], but it functions as a binding target of exosomes was firstly demonstrated in this study.

Several proteins or peptides have been reported to be ligands of FPRs, like fMLF[42], Annexin A1[43], LL-37[44], FAM3D[23], uPAR$_{88-95}$[45], which are synthetic peptide agonists and nonpeptide agonist. All these molecules are soluble components that chemoattract or activate target cells via FPRs. In our study, extracellular SCIMP peptides were proven to chemoattract and stimulate neutrophils via FPRs, but naturally cleaved SCIMP extracellular peptides have not been described. Therefore, the interaction of SCIMP with FPRs in vivo should depend on exosome secretion.

**Fig. 5 | SCIMP can elevate the survival rate of the *E. coli*-induced ALI model by chemoattracting more neutrophils to lung. a** Two schematic workflows show the processes to explore the effect of exosome inhibitor on the ALI model and the effect of murine SCIMP$^N$/SCIMP$^+$ exosome on the ALI model (i.p: intraperitoneal injection; b.p: bronchial perfusion). **b**–**d** After *C57* mice were pretreated with the exosome inhibitor GW4869 (50 μg per mouse) or vehicle for 1 h, *E. coli* ($1 \times 10^7$ CFU per mouse) was perfused into the lung; The pulmonary tissue injury stained with H.E (bar = 200 μm), the SCIMP$^+$ exosomes in the BALF, and the survival rate of the ALI model were detected (*n* = 6 in each group). **e**, **f** Mice were perfused with the *E. coli* ($1 \times 10^7$ CFU per mouse) accompanied by murine SCIMP$^N$ (10 μg per mouse), murine full-length SCIMP$^{pro}$ (100 μg per mouse), the vehicle of peptides/protein (PBS, 50 μL per mouse), murine SCIMP$^+$ exosome (10 mg in 50 μL PBS per mouse), or SCIMP$^-$ exosomes (10 mg in 50 μL PBS per mouse), and the effect of SCIMP$^N$,

SCIMP$^{pro}$, and SCIMP$^{exo}$ on the lung injury and survival rate of ALI model were evaluated (bar = 200 μm, *n* = 10 in each group). **g**, **h** Neutrophils (CD11b$^+$Ly6G$^+$F4/80$^-$) in the BALF of *E. coli* ($1 \times 10^7$ CFU per mouse) induced ALI model (*n* = 8 in each group) treated with the vehicle buffer (PBS, 50 μL per mouse), SCIMP$^N$ (50 μg per mouse), SCIMP$^-$ exosomes from CHO cells (10 mg in 50 μL PBS per mouse), or murine SCIMP$^+$ exosomes (10 mg in 50 μL PBS per mouse), respectively, at 4 h of the model were detected by cytometry. **i**, **j** The *E. coli* clone counts in the BALF of *E. coli*-induced ALI model (*n* = 6 in each group) treated with the vehicle buffer (PBS, 50 μL per mouse), SCIMP$^N$ (50 μg per mouse), SCIMP-negative exosomes from CHO cells (10 mg in 50 μL PBS per mouse), and murine SCIMP$^+$ exosomes (10 mg in 50 μL PBS per mouse), respectively, at 4 h of the model were determined. The raw data is available in the "Source Data".

## Methods

### Ethics statement

In this study, all animal experiments, clinical sample collection, clinical information gathering, and clinical sample processing were approved by the Ethics Committee of Blood Diseases Hospital, Chinese Academy of Medical Sciences (Approval No. CAMSCRF2021013-EC-2), and the Ethics Committee of Tianjin First Central Hospital (Approval No. 2019N134KY) according to the Declaration of Helsinki. Informed consent was obtained from all human research participants.

### Clinical samples collection

A total of 57 peripheral blood specimens from patients with hematological diseases and pulmonary infection, and 50 peripheral blood specimens from patients with hematological diseases without pneumonia or any other infection were included in this study. A total 262 time-pointed specimens were regularly collected along with the monitoring on hemogram (usually 2–3 times/week) or especially collected on the days when the patient developed fever or other infectious symptoms/signs. The number of samples taken after the day when the patient developed fever or other infectious symptoms/signs was the same as the number of samples taken prior to that day. In addition, peripheral blood specimens from 10 healthy volunteers were enrolled. Healthy volunteers, of whom the median age was 32 (17–47) years, were without any underlying diseases or abnormal results in blood routine blood tests, liver and kidney function tests, routine urine tests, routine stool tests (occult blood), infection-related markers tests, chest X-ray, etc. Bronchoalveolar lavage fluid, totaling 56 specimens, were collected from patients accepting fiberoptic bronchoscopy due to pneumonia or pulmonary neoplasms.

### Reagents and mice

The peptides in this study were biosynthesized by the Chinese-Peptide-Company.Inc and the sequences and immune epitope predictions of these peptides are available at Supplementary Fig. S15. The SCIMP protein and SCIMP-containing exosomes were purified according to the methods described below. The HEK293, CHO and RAW264.7 cell lines were purchased from the ATCC.

The antibodies, including anti-His antibody (MA1-21315-HRP, Invitrogen), anti-CD9 antibody (VJ1/20, Abnova), anti-CD63 antibody (353039, BioLegend), anti-CD81 antibody (abs132701, Absin), anti-TSG101 antibody (ab30871, Abcam), anti-EEA1 antibody (3288T, CST), anti-LAMP2A antibody (sc-20011, Santa Cruz), PerCP-Cy5.5-anti-CD11b antibody (101228, BioLegend), APC-anti-Ly6G antibody (127614, BioLegend), PE-anti-CD11c antibody (117308, BioLegend), PE-Cy7 anti-F4/80 antibody (123114, BioLegend), FITC-anti-FPR antibody (W15086B, BioLegend), and anti-CD9 capture beads (ab239685, Abcam), were commercially purchased. The chemicals used in this study, including LPS (L2630, Sigma), CSH (SML1575, Sigma), DHR probe (HY-101894, MCE), GW4869 (HY-19363, MCE), fMLF (F3506, Sigma), IL-8 (ab259397, Abcam), Ca$^{2+}$ probe (F14218, Invitrogen) and BODIPY FL iodoacetamide

(D6003, Invitrogen), were also purchased. The *E. coli* was obtained from ATCC (ATCC#19138).

*C57BL/6* mice were purchased from Beijing Vitalstar Bio-technology Co., Ltd., and *C57BL/6J-SCIMP$^{-/-}$* mice were generated by Shanghai Model Organisms Center, Inc. The mice were housed in a special pathogen-free (SPF) facility at the Institute of Hematology & Blood Diseases Hospital, Chinese Academy of Medical Sciences & Peking Union Medical College. All procedures were approved and monitored by the Animal Care and Use Committee of the Institute of Hematology & Blood Diseases Hospital, Chinese Academy of Medical Sciences.

### Cell culture and transfection

The HEK293, L929, BMDM, and RAW264.7 cell lines were cultured in medium containing 10% Fetal Bovine Serum (FBS, Gibco) in Dulbecco's Modified Eagle Medium (DMEM, Gibco) with 1×Penicillin/Streptomycin (Gibco), and at 5% $CO_2$ and 37 °C. The subculture methods of these cell lines followed the recommendations from ATCC. To transfect the plasmid into HEK293 cells, 5 μg of purified plasmid was added to 200 μL of polyethylenimine (PEI, 1 μg/mL in Opti-MEM, Sigma) for 15 min at room temperature. Then, the mixture was added dropwise into the supernatant of HEK293 cells at 70–80% confluence, and 8 h later the supernatant was replaced with the fresh medium.

### Preparation of SCIMP$^{pro}$ and SCIMP$^N$

The pET-32a-c(+)-SCIMP-6×his plasmid was transformed into *E. coli* (BL21DE3, from ThermoFisher), and then, BL21DE3 was grown on solid LB plates (Amp resistance). Colonies were picked and cultured in liquid LB (Amp resistance), and when the OD600 of the bacterial medium exceeded 0.8, 1 mM IPTG was added to the medium and the culture continued at 37 °C and 300 rpm overnight. After collecting the pellet of SCIMP-6×his-overexpressing BL21DE3 cells by centrifuge, the pellet was lysed using lysozyme and sonication. Following centrifugation of the cell lysate, the supernatant containing SCIMP-6×his was collected and filtered through a 0.45-μm filter membrane. During the purification of SCIMP$^{pro}$ with a Ni-NTA column, in addition to using 500 mM sodium chloride, the imidazole concentration in the washing buffer was 100 mM, and that in the elution buffer was 500 mM. The purity of SCIMP$^{pro}$ was evaluated by SDS-PAGE, stained with Coomassie light blue, quantified using Bovine Serum Albumin (BSA), and stored at −80 °C until use.

The human and murine SCIMP$^N$ were biosynthesized as mentioned above, each with a purity of more than 95%. Their amino acid sequences were "MDTFTVQDSTAMSWWRNN" and "MSWWRDNF", respectively.

### The preparation of SCIMP antibody

The SCIMP antibody was generated by immunizing rabbits with purified prokaryotic SCIMP protein, and the purity of the SCIMP protein is shown in Supplementary Fig. S16. After obtaining SCIMP-immunized

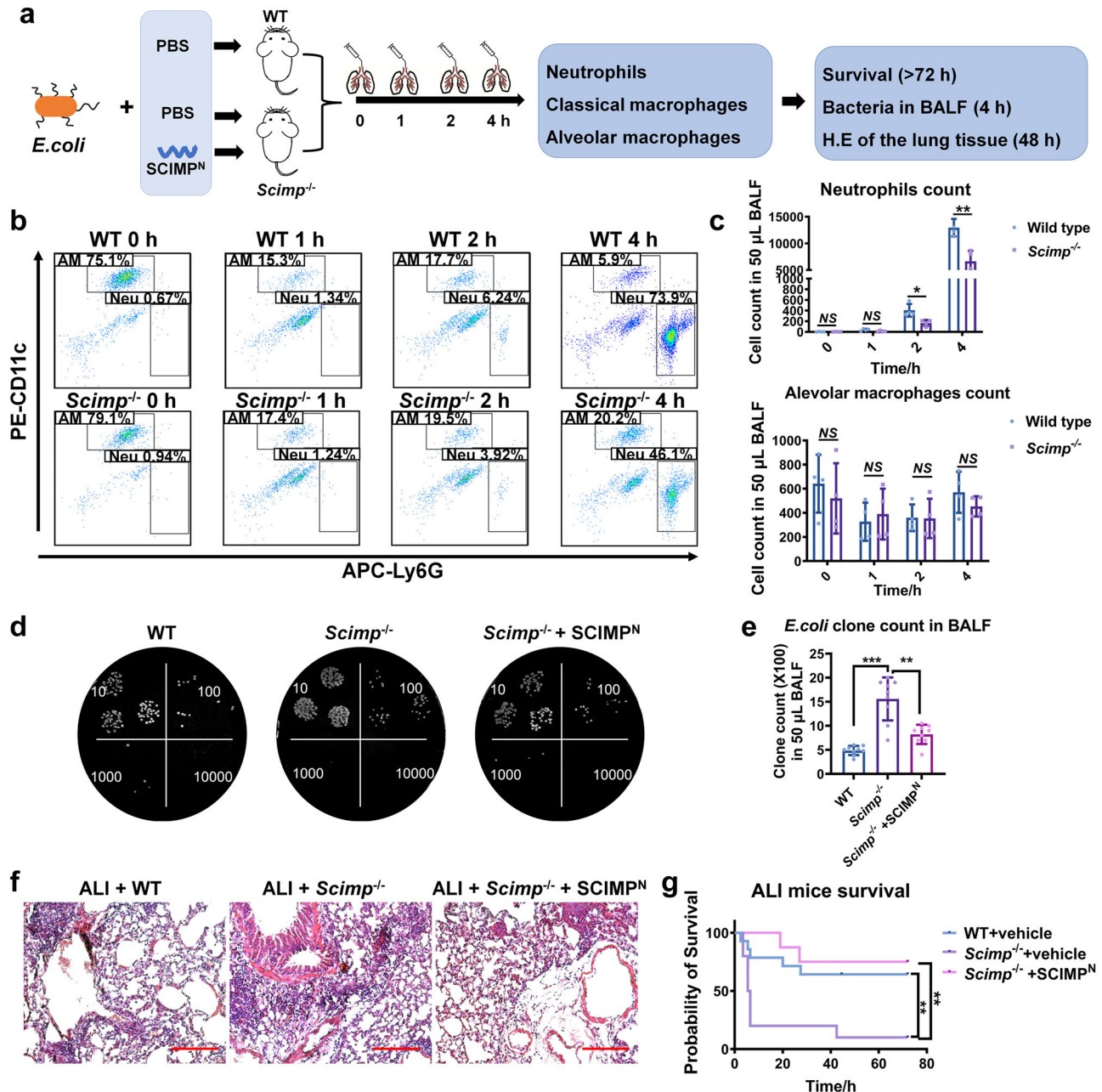

**Fig. 6 | SCIMP plays a key role in neutrophil chemotaxis, bacterial clearance, and survival in the ALI model. a** The schematic workflow shows the study of immune response and survival of the ALI model in WT and *Scimp*[-/-] mice. After perfusing WT and *Scimp*[-/-] mice with *E. coli* suspension ($5 \times 10^6$ CFU per mouse) respectively along with the murine SCIMP[N] (50 μg per mouse) or the vehicle buffer (PBS, 50 μL per mouse), the survival rates of the three groups were observed and analyzed. **b**, **c** The count of neutrophil (Neu, CD11b[+]Ly6G[+]F4/80[-]) and alveolar macrophage (AM, CD11b[+]CD11c[+]F4/80[+]) in the BALF of *E. coli*-induced ALI model in WT mice (*n* = 6) and *Scimp*[-/-] mice (*n* = 6) were measured by cytometry at 0, 1, 2, and 4 h (two-sided *t*-test). **d**, **e** The bacterial clone count at 4 h of *E. coli*-induced ALI model in WT mice (*n* = 6) and *Scimp*[-/-] mice (*n* = 6) with or without rescued by murine SCIMP[N] (50 μg per mouse) were determined and statistically calculated by diluted bacterial culture. **f**, **g** The pulmonary tissue injury and inflammatory cell infiltration observed under the H.E staining biopsies (bar = 200 μm) and the survival of *E. coli*-induced ALI model of the WT and *Scimp*[-/-] mice with or without rescued by bronchus-perfused murine SCIMP[N] (50 μg per mouse) were evaluated. The raw data is available in the "Source Data".

rabbit serum, we evaluated the titer of the antibody in the serum that could recognize the SCIMP protein using directed binding ELISA. Protein A/G-conjugated beads were employed to extract the total antibody from the serum, and then the SCIMP-sourced peptides (listed in Supplementary Fig. S15) were conjugated to CNBr-activated Sepharose 4B and used to accumulate peptides-recognizing antibody. The ability of purified antibody was assessed by recognizing the SCIMP-containing bacteria lysate or cell lysate (Supplementary Fig. S17 and Fig. 1c) and utilized in the subsequent experiments.

## Exosome extraction methods
Cell culture supernatant, serum and BALF samples were collected and centrifuged to remove cells and debris as previously reported[46]. In brief, the acquired supernatants were then ultracentrifuged at $100,000 \times g$ for 70 min at 4 °C. After washing with an equal volume of PBS twice (ultracentrifuged at $100,000 \times g$ for 70 min at 4 °C), the exosome pellet was finally resuspended in PBS, and stored at −80 °C. Serum and BALF samples were collected and centrifuged as mentioned above and the exosomes in serum or BALF samples were purified by

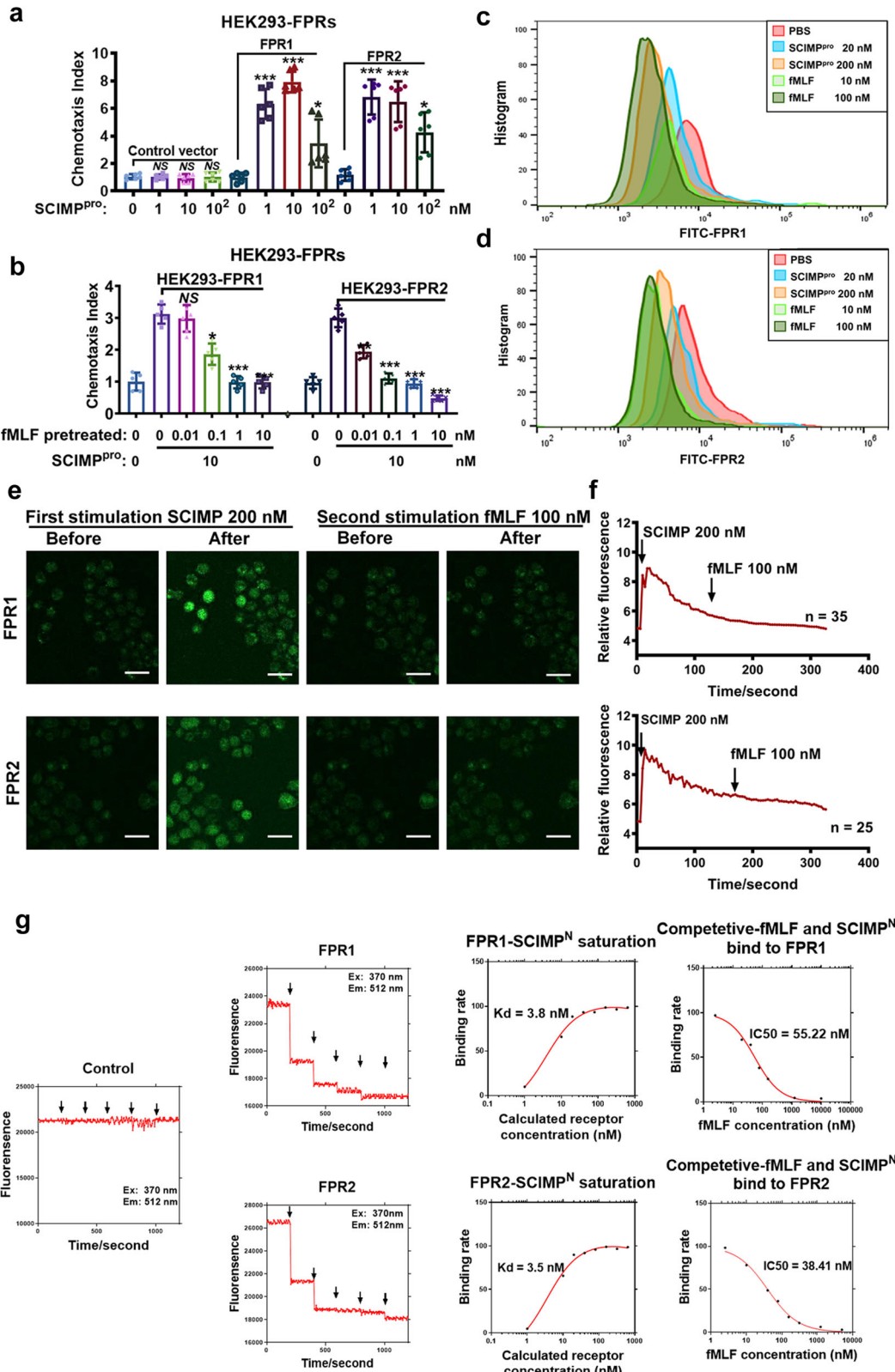

following the manufacturer's protocol (UR52151, UR52161, Umibio Science and Technology Group). In brief, the exosome concentration solution was added to the serum or BALF samples, and then, the samples were completely mixed and incubated at 4 °C overnight. The samples were centrifuged at 12,000 × g for 2 min at 4 °C, and the pellets were resuspended in PBS and transferred into the upper chamber of the SEC columns. After centrifugation at 3000 g at 4 °C for

10 min, the purified exosomes were eluted from the SEC columns and stored at −80 °C.

**Immunofluorescence (IF) and immunohistochemistry (IHC)**
IF: RAW264.7 cells that stably expressed exogenous SCIMP-GFP fusion protein or GFP were cultured on sterile glass slides in 12-well plates for 24 h and then heated-denatured *E. coli* (ATCC#19138, MOI = 100), LPS

**Fig. 7 | The FPR1/2 are identified as the receptors of SCIMP with a high affinity. a** The chemotactic activity of SCIMP[pro] (0, 1, 10, 100 nM) to FPR1/2 was determined by the Boyden Chamber chemotactic assay using the FPR1/2-HEK293 cells or the HEK293 cells transfected with the empty plasmid ($n = 5$ in each group). **b** After preincubating the cells with fMLF at different concentrations (0, 0.01, 0.1, 1, 10 nM) to desensitize the FPRs for 1 h, the chemotaxis activity of SCIMP[pro] (10 nM) to HEK293 cell lines expressing FPR1 or FPR2 was measured by the Boyden Chamber chemotaxis assay ($n = 6$ in each group). **c, d** After incubating FPR1/2-HEK293 cells with fMLF (10, 100 nM) and SCIMP[pro] (20, 200 nM) for 30 min, the distribution of FPR1/FPR2 on the cell membrane of these HEK293 cells was detected by cytometry with FITC-anti-FPR antibodies. **e, f** After FPR1/2-HEK293 cells were loaded with the

Ca[2+] probe, the time-coursed calcium influx in the cells stimulated by SCIMP[pro] (200 nM) for 180 s and then stimulated by fMLF (100 nM) subsequently was recorded under the confocal microscopy (scale bar = 50 μm). **g** After labeled with BODIPY FL Iodoacetamide, the SCIMP[N] (final concentration: 1, 10, 20, 40, 80, 160, 320, 640 nM) were incubated with a fixed amount of membrane component of FPR1/2-HEK293 cells (total protein amount: 100 mg per well), and the fluorescence density at the emission wavelength (512 nm) was measured, as well as the fMLF (5, 10, 20, 40, 80, 160, 320, 640, 1280 nM) to competitive binding to the dye labeled SCIMP[N] (10 μM) and FPR1/2 (total protein amount: 100 mg per well). The raw data is available in the "Source Data".

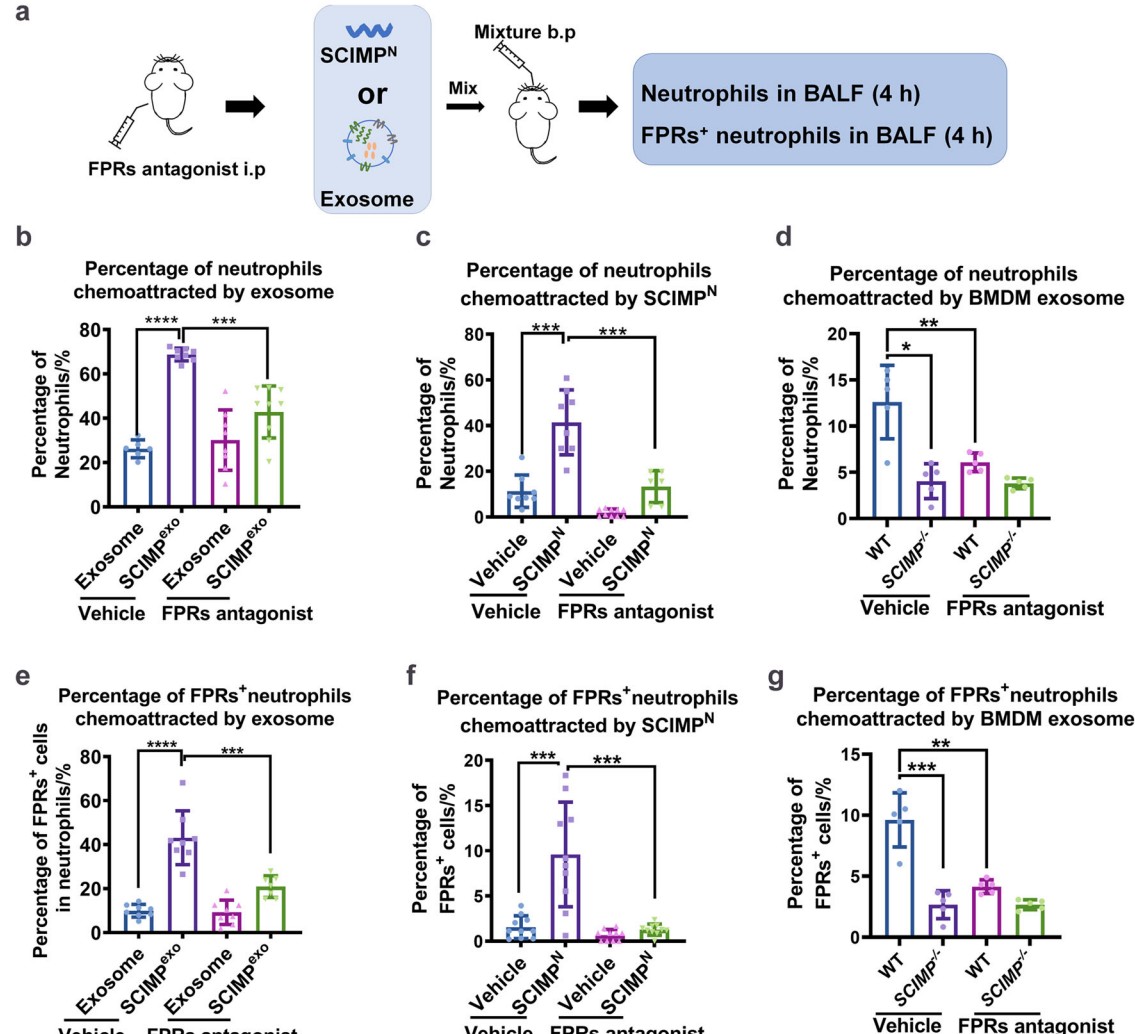

**Fig. 8 | The FPRs antagonist could inhibit the chemotaxis of SCIMP to neutrophils in vivo. a** The schematic workflow shows that pretreatment with FPRs antagonist (CsH, 50 μg per mouse) for 1 h can inhibit the chemotaxis of bronchus-perfused murine SCIMP[N] (50 μg per mouse), exogenous murine SCIMP[exo] (10 mg per mouse), and endogenous murine exosomes (10 mg per mouse) to neutrophils in the pulmonary situ. **b, e** The chemotaxis activity of murine SCIMP[exo] (10 mg per mouse) from the supernatant of SCIMP-overexpressing CHO cells and the control exosomes (from the supernatant of empty vector transfected CHO cells, 10 mg per mouse) to neutrophils or FPRs[+] neutrophils in the lung of *C57* mice was assessed

with or without pretreatment of FPRs antagonist using flow cytometry ($n = 6$ in each group). **c, f** The chemotaxis activity of the murine SCIMP[N] (50 μg in 50 μL per mouse) and the vehicle (PBS, 50 μL per mouse) to neutrophils or FPRs[+] neutrophils in the lung of *C57* mice with or without pretreatment of FPRs antagonist was measured by flow cytometry ($n = 9$ in each group). **d, g** The chemotaxis activity of the ultracentrifuge purified endogenous exosomes released by WT or *Scimp[-/-]* BMDMs to neutrophils or FPRs[+] neutrophils in the lung of *C57* mice with or without pretreatment with FPRs antagonist was measured by flow cytometry ($n = 5$ in each group). The raw data is available in the "Source Data".

(1 μg/mL), or PBS was added to the supernatant and incubated for 0, 1, and 4 h. The cells on slides were fixed in 4% paraformaldehyde for 15 min and permeabilized in 0.5% Triton X-100 for 20 min at room temperature. Next, the cells on the slides were blocked with 1% BSA, followed by incubation with specific primary antibodies (anti-LAMP2A and anti-EEA1) and then the fluorescent secondary antibodies. The

distribution of SCIMP, LAMP2A and EEA1 in the cell was observed by the confocal microscope (Andor dragonfly, Zeiss).

IHC: Lung biopsies of mice that were infected with *E. coli* were fixed with paraffin. The paraffin sections were processed according to a previously described method[22] and stained with the SCIMP antibody (1 μg/mL) in PBS.

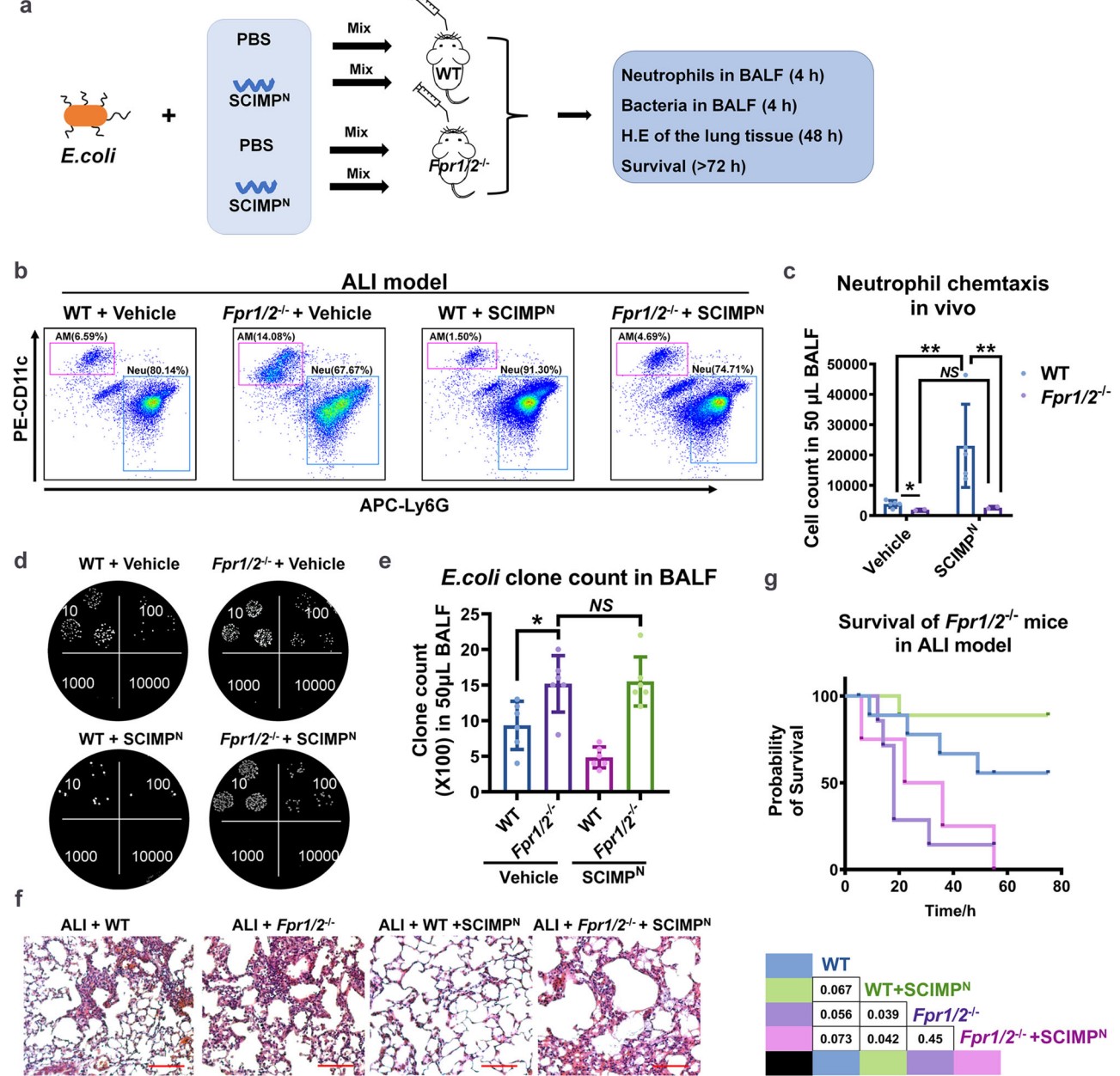

**Fig. 9 | FPRs expressed on neutrophils are necessary for the effect of SCIMP on the ALI model. a** The schematic workflow shows the study of the effect of murine SCIMP^N on the ALI model by using WT or *Fpr1/2^-/-* mice. **b–g** After transbronchially perfused with SCIMP^N (50 μg/50 μL PBS per mouse mixed with $5 \times 10^6$ CFU *E. coli*, *n* = 5) or Vehicle (50 μL PBS per mouse mixed with $5 \times 10^6$ CFU *E. coli*, *n* = 5) for 4 h to WT or *Fpr1/2^-/-* mice, neutrophils (Neu, CD11b⁺CD11c⁻Ly6G⁺) and alveolar macrophages (AM, CD11b⁺CD11c⁺F4/80⁻Ly6G⁻) in the BALF from the ALI model were measured by flow cytometry (**b**, **c**), and the bacterial count was measured by dilutional bacterial clone culture (**d**, **e**, *n* = 5), and the lung injury and inflammation were evaluated by histological examination of lung biopsies stained with H.E. (bar = 200 μm, **f**), and the survival of the four groups (*n* = 8 in each group) was observed and analyzed (**g**). The raw data is available in the "Source Data".

## Transmission electron microscopy (TEM)

TEM Glow-discharged copper grids were immersed in a suspension of exosomes for 20 min and fixed with 4% paraformaldehyde for 1 min. After blocking with 5% BSA for 10 min, the grids were incubated with specific primary antibodies (SCIMP antibody, 1 μg/mL) and gold-conjugated secondary antibodies (1:1000, G7277, Sigma). Then, they were treated with 1% glutaraldehyde for 2 min, followed by negative staining with 2% uranyl acetate, and examined with an HT7800 transmission electron microscope (Hitachi, Japan).

## Exosomal SCIMP detection system

Plasma, body fluids or cell culture supernatants were repeatedly centrifuged to remove the cells, dead cells, or cell debris as mentioned above. Fifty microliters of pretreated sample + 50 μL of sterile PBS were mixed well before incubation with CD9 Capture Beads (50 μL/sample) at 4 °C overnight. The next day, the samples were incubated with the FITC-conjugated SCIMP antibody (1 μg/mL, 1:50), PE-anti-CD63 antibody (1:100) at 37 °C for 60 min. All samples were analyzed by a FACS Canto II (BD, USA) following the gating strategy (Fig. 2b).

## Single extracellular vesicle (EV) detection assay (nanoparticle flow cytometry)

The EVs in the serum or BALF samples were extracted and purified by the SEC column (UR52151, UR52161, Umibio Science and Technology Group). All PBS and antibody solutions were filtered through a 50 nm

syringe filter to exclude nonspecific nanoparticle contamination. The antibodies used in this assay were diluted 100 times relative to the concentration that was used for cytometry. After incubated with the antibodies, the purified exosomes were analyzed by a CytoFlex-S (Beckman) according to a previously described protocol[47].

### E. coli- and LPS-induced ALI models

The *C57BL/6* mice, aged 6–8 weeks, weighing 20–25 g, were randomly grouped. The mice were anaesthetized with tribromoethanol (Avertin, 200 mg/kg, i.p) and depilated on the front of the neck. Then, they were perfused with 50 μL of bacterial suspension ($5 \times 10^6$ CFU *E. coli*, ATCC#19138) + 50 μL of air through the trachea. The survival rate of the models perfused with different doses of *E. coli* or LPS was measured. The surgical incision was sutured with a skin stapler, and the mice were placed on an electric blanket at a warm temperature to recover. The endpoint of the ALI model was the inability to ambulate or maintain an upright position that prevents the animal's easy access to food and/or water for 12 h. In the experiments involving ALI models, the mice were male.

### Chemokine and receptor interaction assay

Chemotaxis assays were conducted using the Body Chamber and TaxiScan systems to assess cell chemotactic activity. Genetically modified HEK293 cells, PBMC cells, or neutrophils were placed in the upper chamber of the chemotaxis chamber ($1 \times 10^4$ cells in 45 μL DMEM) or in the left well of TaxiScan ($1 \times 10^3$ cells in 10 μL DMEM). Various concentrations of chemoattractants, such as exosomes, proteins, or peptides, were added to the corresponding lower chamber or right well. The Body Chamber's upper and lower layers were separated by cellulose acetate membranes with specific pore sizes to allow cells attracted by chemotactic factors to deform and migrate through. Finally, cell staining was used to detect the number of cells adhering below the central membrane and in the lower well. The TaxiScan system had a channel between its left and right wells, and we recorded in real-time, under a microscope, the radial velocity of cells migrating toward in response to chemotactic stimuli.

Receptor internalization assays were conducted in HEK293 cells overexpressing FPR1/2 or FPR1/2-GFP. After co-incubating cells with different concentrations of stimuli for a specific time, FPR1/2 antibodies were used to label FPR1/2-HEK293 cells, and flow cytometry was employed to detect the expression levels of FPR1/2 on the cell membrane. Immunofluorescence (IF) confocal microscopy was used to examine the internalization of membrane-surface receptors in FPR1/2-GFP-HEK293 cells.

Calcium flux assays were performed using calcium probe-loaded HEK293 or FPR1/2-HEK293 cells. Various doses of stimuli were added, and the fluorescence intensity of the probe after intracellular calcium ion release was observed in real-time using laser confocal microscopy and analyzed by ImageJ.

### Ligand and receptor binding assay

SCIMP[N] peptides were labeled with BODIPY FL iodoacetamide according to the manufacturer's instructions. Based on the dimer quenching characteristic (quenching when the distance of molecules was less than 2 nm[48] of BODIPY FL iodoacetamide, the free BODIPY FL iodoacetamide labeled SCIMP[N] should quench when bond to the receptors on the membrane debris from the HEK293 overexpressing FPRs. In this experiment, we used a fixed amount of BODIPY FL iodoacetamide labeled SCIMP[N] and the variable doses of the competitive agents and FPRs proteins. In the black/clear bottom 96-well plate, the interactions of SCIMP[N] with human FPR1, human FPR2, murine FPR1 or murine FPR2 protein were measured by using the Synergy H4 microplate reader (BIO-TEK, USA). Labeled SCIMP was excited at 370 nm and detected at 512 nm. The kinetic analysis was titrated using labeled SCIMP with unlabeled human FPR1, human FPR2, murine FPR1

or murine FPR2. Moreover, the competing kinetic analysis was titrated, but unlabeled FPR1, FPR2, murine FPR1 or murine FPR2 were incubated with serially diluted fMLF in advance.

The association/dissociation constant of ligand to receptor was displayed by "Kd". In the binding assay, the free labeled SCIMP[N] without FPRs protein displayed the highest fluorescence intensity, because no quenching occurred. When different doses of FPRs solution (diluted the FPRs containing cell membrane debris by 1 to 100,000 times) were added into the wells with free labeled SCIMP[N], the fluorescence intensity was measured for 4 min. The well in which the fluorescence intensity was no longer reduced anymore by the higher concentration of FPRs, and this well would be regarded as the saturation point. Since the binding reaction pattern of GPCR to ligand is "[A] + [B] = [AB]", the concentration of FPRs at the saturation point was calculated. And the association constant (Kd) of SCIMP[N] to FPRs according to the saturation curve. In the competitive assay, the free labeled SCIMP[N] were mixed with various concentrations of fMLF, and then incubated with fixed concentration of FPRs, and the fluorescence intensity was measured for 4 min. Since the dissociation reaction pattern of GPCR to ligand is "[AB] + [C] = [AC] + [B]", the middle fluorescence intensity corresponding to the IC50 was calculated.

### In situ chemotaxis assay

Briefly, mice aged 6–8 weeks and weighing 20–25 g were randomly grouped, ensuring a gender ratio of 1:1 between males and females in each group. The mice were then administered SCIMP[exo], the control exosomes, or SCIMP[N] through the trachea as mentioned above. Five hours later, bronchoalveolar lavage fluid (BALF) was collected by perfusing the lung with 1 mL of PBS for three times. Cells in the BALF were labeled with the antibodies (PerCP-Cy5.5-CD11b, APC-Ly6G, PE-CD11c, PE-Cy7-F4/80, FITC-FPR1/FPR2). All samples were analyzed by a Cytoflex-S (Beckman, USA).

### Neutrophil function assay

First, human peripheral blood mononuclear cells (PBMCs) and neutrophils (PMNs) were isolated by density gradient centrifugation as previously described[26]. The chemotaxis assay of PMNs was measured using the TAXIscan system (TAXIscan FL ECI, Japan) and Boyden Chamber Assay (AP48, Neuro Probe, Inc.), and the chemoattractants were SCIMP[exo], SCIMP[pro], SCIMP[N], fMLF, IL-8 or SDF-1. The bacterial killing assay was carried out in the *E. coli*-induced ALI model and BALF samples were diluted and dripped onto solid LB plates. After 24 h, the numbers of colonies were counted and analyzed.

### Statistics and reproducibility

In this study, the two-sided Student's *t*-test was used to determine the statistical significance of differences in mean values. The data are presented in all figures as mean values ± SEM. A *p* value of <0.05 was considered statistically significant. The "NS", "*", "**", "***" and "****" means the *p* value of two-sided *t*-test ">0.05", "0.01-0.05", "0.001-0.01", "0.0001-0.001", and "<0.0001". The biological replicates of each group in the in vitro and in vivo experiments of this study were usually no less than five. The experiments in this study were independently repeated for no less than three times. The raw data used for statistical analysis in this study are available in the "Source Data".

### Reporting summary

Further information on research design is available in the Nature Portfolio Reporting Summary linked to this article.

## Data availability

The raw data used for the graph and the original image of the western blot in this paper are available as "Source Data." The raw data used for the supplementary figures in this study are also available as "Source Data." Additionally, the raw data of the proteomic analysis are

accessible as "Source Data" and have been deposited in the ProteomeXchange (PXD046185). If there are any questions on the data, method, study design or materials requirement, please contact peixiaolei@ihcams.ac.cn Source Data are provided with this paper.

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

## Acknowledgements

National Natural Science Foundation of China (81700099, 82072046 and 82271884), Science and Technology Project of Tianjin (21JCZDJC01170), Chinese Academy of Medical Sciences Innovation Fund for Medical Sciences (2021-I2M-1-017; 2021-I2M-C&T-B-080), and Haihe Laboratory of Cell Ecosystem Innovation Fund (22HHXBSS00036). Natural Science Foundation of Beijing Municipality (7232089). We are grateful to Professor Dalong Ma (Department of Immunology, School of Basic Medical Sciences, NHC Key Laboratory of Medical Immunology and Center for Human Disease Genomics, Peking University) for the omics strategies and valuable suggestions.

## Author contributions

S.F., Y.W., and X.P. conceived the study and designed all experiments. X.P., L.L. (affiliations 1, and 2), J.W., and D.Z. performed the experiments. Q.R., R.M., C.G., J.W., Q.L., L.L. (affiliation 4), J.L., X.F., Y. Zhang, Y. Zheng, and P.W. collected the clinical samples and provided the technical support. P.J. and E.J. provided the clinical information of the samples. S.F., Y.W. and X.P. wrote the manuscript. All experiments were supervised by Y.W. and S.F.

## Competing interests

The authors declare no competing interests.
