## [Peer Review File · Nature Communications]

REVIEWER COMMENTS

Reviewer #1 (Remarks to the Author):

SCIMP is a transmembrane adaptor protein that is involved in signaling through MHCII glycoproteins and pattern recognition receptors, including TLRs and Dectin-1. In this manuscript, the authors propose a novel and somewhat unexpected role for this protein and exosomes containing this protein as ligands for chemotactic receptors of the FPR family. They support this claim with multiple lines of evidence. In particular, they show that:

1. SCIMP is present in exosomes from SCIMP-expressing cells and in exosomes isolated from BALF of patients with pneumonia.
2. exosomes from SCIMP-expressing cells and isolated SCIMP protein attract neutrophils in vitro in a migration assay and in vivo into mouse lungs
3. The N-terminus of SCIMP and FPR family receptors are critical for this chemoattraction.
4. SCIMP peptides and SCIMP-containing exosomes improve the outcome of acute lung injury in a mouse model, whereas SCIMP and FPR deficiency have the opposite effect.

The study is novel, interesting and very thorough. It addresses the problem from multiple angles and it is largely convincing. However, it also has weaknesses and shortcomings, which have to be addressed:

1. Authors generated a new anti-SCIMP polyclonal antibody. However, no data on its specificity and functionality are provided. Authors should describe the method of SCIMP antibody generation, what exactly was used as immunogen, mainly which part of SCIMP protein was used, what was its source, and if and how the antibody was purified. Since it is a new antibody, they should also provide data demonstrating its specificity. In Fig. 1F (in the whole blots provided as supporting material) the anti-SCIMP antibody stains many proteins (essentially the entire lanes) in E. coli treated samples. This raises the suspicion that this polyclonal antibody was raised against recombinant SCIMP isolated from bacteria and may contain antibodies to contaminating bacterial proteins. Authors should provide evidence that their antibody does not stain E.coli proteins since it is used on multiple occasions on E.coli-treated samples. This problem could potentially affect data in Fig. 1F, 1H, and multiple panels in Fig. 2. For Fig. 2 verification of the antibody specificity is especially critical. As an additional measure to make SCIMP detection data more convincing it is essential to show empty vector-transfected controls in Fig.1C,D,E or SCIMP^{-/-} mice for Fig.1 H.

2. The data in Figure 1 proving that SCIMP is part of the exosomes are not sufficiently convincing. Western blot quality in Fig.1C is rather low. For exosome markers CD63 and CD81, the authors seem to have picked bands of corresponding molecular weight among many others and it is not very clear if they

indeed represent these antigens. In Fig.1F as mentioned above, the SCIMP antibody stains almost the entire lanes, which could reflect reactivity with multiple bacterial proteins. Alternative explanation is that SCIMP is present in some poorly soluble aggregates, incompletely dissolved in SDS. This would then go against the hypothesis that SCIMP is present in exosome membranes and rather suggest that it is in some protein aggregates coming e.g. from dead cells. In the TEM data in Fig 1E, the dark areas could also represent protein aggregates. Flow cytometry data throughout the paper are more convincing as well as mass spectrometry data in Fig.3, but the abovementioned data in Fig. 1 must be improved. In addition, authors used two methods of exosome purification. Differential centrifugation and a kit of unclear origin. Authors should clearly state the type of the kit and the manufacturer and indicate in which experiments they used the kit and where they used differential centrifugation and how the purity of exosomes was assessed in both cases.

3. In Fig. 1I, EEA1 is also an early endosome marker, LAMP2A is late endosome/lysosome marker. Their co-localization with SCIMP could have multiple interpretations. Authors should comment on this in the manuscript and provide reference documenting usability of these proteins as exosome secretion markers.

4. Regarding the SCIMP topology in exosomal membrane, SCIMP N-terminus is extremely short and so it is surprising that it is accessible to antibody. Authors should test whether the C-terminus can also be detected on the surface of these exosomes (e.g. with anti His-tag antibody used in Fig. 1B). Blocking the antibody staining with SCIMP N-terminal peptide in Fig S5 potentially makes a strong argument. However, adhering the exosomes on plastic during ELISA could lead to their damage and membrane topology disruption. It would be more convincing to demonstrate the ability of SCIMPN peptide to block SCIMP antibody binding to exosomes in a flow cytometry based NTF assay. Authors should also verify if SCIMP with truncated N-terminus used in Fig. 4C is still expressed and targeted to cellular membranes/exosomes.

5. Potential shortcoming of the data on FPR receptor activation by SCIMP N-terminus is the fact that for experiments demonstrating FPR involvement in Fig.7A-E, SCIMPpro was purified from bacteria and therefore would have formylated N-terminus – a prototypic FPR ligand. Its preparation could also be contaminated by additional bacterial proteins activating FPRs. In addition, Methods section seems to suggest that SCIMPN peptide was also prepared in bacteria. Authors have to clarify if SCIMPN was synthetic peptide or produced in bacteria. Bacterial origin of SCIMPN would severely compromise majority of FPR-related data in the manuscript. Finally, in Fig7A-E, negative control, i.e. cells not expressing FPRs, should be shown. The sequence of SCIMPN peptide should also be provided.

6. Experiments in Fig. 7G should be explained more clearly, especially arrows in the left panel and K_d calculation method

7. It is important to show that endogenous levels of SCIMP in endosomes are able to attract neutrophils to lungs. This authors attempted to do in Fig. 8D, G. However, there they used exosomes from E.coli treated Raw cells which could be contaminated with bacteria. Using the exosomes from SCIMP^{-/-} and WT bone marrow derived macrophages would provide necessary controls missing in the current Fig. 8D, G.

8. Potential weakness of this manuscript is the design of ALI mouse model, where massive E.coli infection leads to death within hours. Here, higher neutrophil infiltration is useful to combat the high bacterial load. It is a good evidence of the ability of SCIMP exosomes to attract neutrophils and have impact in vivo. On the other hand, it is unclear how exactly is this situation is relevant for human patients, where the infections develop with slower kinetics and more neutrophils could also mean more tissue damage. These issues should be properly discussed/explained.

9. SCIMP N-terminus can be expected to be exposed on all SCIMP expressing cells. Is cell-expressed SCIMP accessible to antibody staining? Could it also serve as neutrophil chemoattractant?

10. Human SCIMP has two alternative translation starts generating two different N termini. Do they both have chemotactic properties? How this affects interpretation of the data?

Reviewer #2 (Remarks to the Author):

In the manuscript entitled “Exosomal secreted SCIMP regulates communication between macrophages and neutrophils in pneumonia”, Pei and colleagues present their novel findings on exosomal SCIMP from macrophages after bacterial infection. Functionally, exosomal SCIMP has chemotactic activity and activates neutrophils through FPR1/2. This is a carefully done study with both gain-of-function and loss-of-function approaches. The findings are novel and interesting. However, enthusiasm is dampened by several flaws in the study design and experimental methods.

Major comments:

1. Is SCIMP a secreted protein? As shown in Fig. 1, chemotaxis activity was observed in SCIMP-containing supernatant from SCIMP over-expressing HEK293 cells. Unfortunately, the authors failed to determine if soluble SCIMP, exosomal SCIMP, or both of them contributed to the chemotactic effect. Therefore, the soluble and exosomal SCIMP needs to be quantified using ELISA. In addition, the chemotaxis activity should be carefully compared throughout the work given that soluble SCIMP may play a major role in both in vivo and in vitro conditions.

2. The exosomes are not well characterized. The authors briefly characterized HEK293-derived exosomes. In contrast, exosomes from animals or human subjects were not validated.

3. It is very interesting that SCIMP protein was secreted via exosome from the RAW264.7 cells after stimulation with denatured E.coli but not after stimulation with LPS. Did LPS or bacterial infection induce SCIMP expression at the mRNA level? A previous study has reported that SCIMP promotes selective proinflammatory cytokine responses by direct modulation of TLR4, which is a well-known receptor involved in LPS recognition and signal initiation. In this scenario, it would be better to further investigate the mechanism, although the authors briefly stated in the discussion.

4. The methods and figure legends are not informative. Please describe the details of each method, especially for in vivo experiments. For instance, how the mice were anesthetized? What was used for perfusion? How many mice were used for each experiment?

5. Fig. 5 and 9 indicated that soluble SCIMP (SCIMP peptides) alone has similar effects as exosomal SCIMP. If so, why not focus on soluble SCIMP rather than exosomes? It is not clearly stated.

Minor comments:

1. Please show the dot plot to indicate the sample size.
2. What does “the colocalization of SCIMP and LAMP2A/EEA1 was weak” mean in Figure S1?
3. Several methods were not described, e.g. cell culture, transfection, ELISA, etc.

Point-by-point response to the reviewers' comments:

Reviewer #1 (Remarks to the Author):

SCIMP is a transmembrane adaptor protein that is involved in signaling through MHCII glycoproteins and pattern recognition receptors, including TLRs and Dectin-1. In this manuscript, the authors propose a novel and somewhat unexpected role for this protein and exosomes containing this protein as ligands for chemotactic receptors of the FPR family. They support this claim with multiple lines of evidence. In particular, they show that:

1. SCIMP is present in exosomes from SCIMP-expressing cells and in exosomes isolated from BALF of patients with pneumonia.
2. exosomes from SCIMP-expressing cells and isolated SCIMP protein attract neutrophils in vitro in a migration assay and in vivo into mouse lungs
3. The N-terminus of SCIMP and FPR family receptors are critical for this chemoattraction.
4. SCIMP peptides and SCIMP-containing exosomes improve the outcome of acute lung injury in a mouse model, whereas SCIMP and FPR deficiency have the opposite effect.

The study is novel, interesting and very thorough. It addresses the problem from multiple angles and it is largely convincing. However, it also has weaknesses and shortcomings, which have to be addressed:

1. Authors generated a new anti-SCIMP polyclonal antibody. However, no data on its specificity and functionality are provided. Authors should describe the method of SCIMP antibody generation, what exactly was used as immunogen, mainly which part of SCIMP protein was used, what was its source, and if and how the antibody was purified. Since it is a new antibody, they should also provide data demonstrating its specificity. In Fig. 1F (in the whole blots provided as supporting material) the anti-SCIMP antibody stains many proteins (essentially the entire lanes) in E. coli treated samples. This raises the suspicion that this polyclonal antibody was raised against recombinant SCIMP isolated from bacteria and may contain antibodies to

contaminating bacterial proteins. Authors should provide evidence that their antibody does not stain E.coli proteins since it is used on multiple occasions on E.coli-treated samples. This problem could potentially affect data in Fig. 1F, 1H, and multiple panels in Fig. 2. For Fig. 2 verification of the antibody specificity is especially critical. As an additional measure to make SCIMP detection data more convincing it is essential to show empty vector-transfected controls in Fig.1C,D,E or SCIMP^{-/-} mice for Fig.1 H.

Response:

Thank you for pointing out these questions. The SCIMP antibody was generated by immune rabbit with purified prokaryotic expressed SCIMP protein, and the purity of the SCIMP protein was shown in **Attached Figure 1**. After we obtained the SCIMP immunized rabbit serum, the Protein A/G conjugated beads was used to extracted the total antibody from the serum, and then the SCIMP sourced peptides (listed in **Attached Figure 2**) were conjugated to the CNBr-activated Sepharose 4B and used to purify the peptides recognizing antibody. The SCIMP specific antibody was evaluated by western blot, as shown in **Attached Figure 3** the SCIMP antibody could recognized the exogenous SCIMP protein expressed by bacteria (BL21DE3) but not the lysate of the bacteria with empty vector transduced. And the purified SCIMP antibody was used in the subsequent experiments.

To double confirm the SCIMP antibody's specificity, we also performed the western blot of exosome free cell supernatant, exosome purified from the cell supernatant and cell lysate of HEK293 cells transiently transfect with the SCIMP-6Xhis vector or empty vector (pCDNA3.1-HIS-B(-)). As shown in **Attached Figure 4**, a band sized at 20 kD could be well stained by SCIMP antibody in the samples of SCIMP overexpressing HEK293 cells' lysate and exosomes, but not those of empty vector transfected cells. Furthermore, the nano particle flow cytometry to detect the SCIMP positive exosome were performed, as shown in the **Attached Figure 5**, the exosome collected from the supernatant of HEK293 transfected with SCIMP-pCDNA3.1 vector could be well labelled with SCIMP antibody, but not the exosome collected from the HEK293 transfected with the empty vectors.

Attached Figure 1

Attached Figure 1. The processes to produce, purify, and evaluate the SCIMP antibody from immunized rabbit serum were schematically shown.

Peptide1#: MDTFTVQDSTAMSWWRNN
 Peptide2#: AKPLKHKQV
 Peptide3#: NESPVQLPPLP
 Peptide4#: QEAPSQPPATYSLVNKVK
 Peptide5#: TVSIPSYI

Attached Figure 2

Attached Figure 2. The peptides sequence from human SCIMP that were predicted to have high immune epitope, and “Peptide1#” is the N terminus of human SCIMP (left); The hydrophilicity of each amino acid in the human SCIMP protein were calculated (right).

Attached Figure 3

Attached Figure 3. The human SCIMP-6Xhis recombinant protein were expressed by the bacteria (BL21DE3) and the SCIMP-6Xhis protein in bacteria lysate were purified by NTA Sepharose and the protein in each fraction during the purification were analyzed by SDS-PAGE and stained with Coomassie blue (**left**); The SCIMP-6Xhis protein in the bacteria lysate samples that expressing SCIMP-6Xhis or transduced with empty vector (pET32a) were stained with the purified SCIMP antibody in Western blot (**right**).

Attached Figure 4

Attached figure 4. The SCIMP protein from the exosome free supernatant, exosome purified from the supernatant (by the ultracentrifuge method), and the cell lysate of HEK293 cells transiently expressing SCIMP-6×his protein or transiently transfected with the empty vector (pcDNA3.1) were stained with the SCIMP antibody.

Attached Figure 5

Attached Figure 5. The exosomes purified by the ultracentrifuge method from the supernatant of HEK293 cells transiently transfected with the empty vector (pcDNA3.1) or expressing SCIMP-6×his protein, were incubated with the CD63 (PE) and SCIMP (FITC) antibodies and analyzed by nano-particle flow cytometry.

2. The data in Figure 1 proving that SCIMP is part of the exosomes are not sufficiently convincing. Western blot quality in Fig.1C is rather low. For exosome markers CD63 and CD81, the authors seem to have picked bands of corresponding molecular weight among many others and it is not very clear if they indeed represent these antigens. In Fig.1F as mentioned above, the SCIMP antibody stains almost the entire lanes, which could reflect reactivity with multiple bacterial proteins. Alternative explanation is that SCIMP is present in some poorly soluble aggregates, incompletely dissolved in SDS. This would then go against the hypothesis that SCIMP is present in exosome membranes and rather suggest that it is in some protein aggregates coming e.g. from dead cells. In the TEM data in Fig 1E, the dark areas could also represent protein aggregates. Flow cytometry data throughout the paper are more convincing as well as mass spectrometry data in Fig.3, but the abovementioned data in Fig. 1 must be improved. In addition, authors used two methods of exosome purification. Differential centrifugation and a kit of unclear origin. Authors should clearly state the type of the kit and the manufacturer and indicate in which experiments they used the kit and where

they used differential centrifugation and how the purity of exosomes was assessed in both cases.

Response:

We thank you for pointing out these questions. We have repeated this experiment, and the exosome in the cell culture supernatant were purified by ultracentrifuge. As show in the attached figure6 and figure1C, along with the exosome free supernatant, purified exosome, and cell lysate, the CD9, CD81, CD63, TSG101, and SCIMP were stained by the respected antibodies on the western blot membrane (**attached figure 5**).

It was surely difficult to distinguish the SCIMP's source in the samples of exosomes purified from the cell culture supernatant or body fluid, due to the defect of the method. Given this, we collected the SCIMP contained cell supernatant of HEK293 cells at 24 hours post the vector transfection in this study, which might have fewer dead cells and cell debris. Additionally, the nano-particle flow cytometry might provide more support the observation of SCIMP existing on the exosomes.

The source of the kit to purify the exosome has been added in the manuscript. By comparing with the exosome purified with two different methods, we observed the different between two exosomes, including the integrity and figure. There also have several research articles reporting the difference of these two methods, and considering the impact on the biological activity of exosome and time consuming, in the experiments that need to evaluate the exosomes activity, we used the ultracentrifuge method to purify exosome, while in the experiments that need to detect the abundancy of exosomes we used the SEC column purification method. In Figure1, we used four antibodies to stain the well-known exosome markers, including CD9, CD81, CD63 and TSG101, as well as SCIMP in the exosome from SCIMP expressing HEK293 cells.

Attached Figure 5

Attached Figure 5. The SCIMP protein and exosome markers (CD9, CD63, CD81, TSG101) from the cell lysate, exosome purified from the supernatant (by the ultracentrifuge method), and the exosome free supernatant of HEK293 cells transiently expressing SCIMP-6×his protein were stained in the Western blot.

3. In Fig. 1I, EEA1 is also an early endosome marker, LAMP2A is late endosome/lysosome marker. Their co-localization with SCIMP could have multiple interpretations. Authors should comment on this in the manuscript and provide reference documenting usability of these proteins as exosome secretion markers.

Response:

We thank you for the helpful suggestions. The EEA1 and LAMP2A has been reported as the markers of the exosome generating process in the cells. Early endosomes, regarded as the main sorting station on endocytic pathway, are characterized by high frequency of homotypic fusions mediated by tethering protein EEA1([doi: 10.1038/s41467-021-24384-2](https://doi.org/10.1038/s41467-021-24384-2)), and EEA1 has been find existing on the membrane of secreted exosome ([doi: 10.1074/mcp.M112.021303](https://doi.org/10.1074/mcp.M112.021303)) and proven to participate in the exosome generation([doi: 10.1038/ncomms4477](https://doi.org/10.1038/ncomms4477)), however, colocalized with EEA1 cannot fully support the hypothesis that SCIMP would enter the exosome generation process. LAMP2A was originally known as the late endosome or lysosome marker, and there has the report that LAMP2A regulates the loading of proteins into exosomes with a ESCRT independent mechanism ([doi: 10.1126/sciadv.abm1140](https://doi.org/10.1126/sciadv.abm1140)). In this study, we used the two proteins colocalizing to SCIMP in the RAW cells under the stimulation of

bacteria and LPS to observe the SCIMP participating in the exosome secreting process, which cannot exclude the possibility that SCIMP does not participate in the exosome generating process solely based on the confocal result. Therefore, we observed the SCIMP protein existing on the membrane of exosome by several methods and verify its biological functions with exosomes.

4. Regarding the SCIMP topology in exosomal membrane, SCIMP N-terminus is extremely short and so it is surprising that it is accessible to antibody. Authors should test whether the C-terminus can also be detected on the surface of these exosomes (e.g. with anti His-tag antibody used in Fig. 1B). Blocking the antibody staining with SCIMP N-terminal peptide in Fig S5 potentially makes a strong argument. However, adhering the exosomes on plastic during ELISA could lead to their damage and membrane topology disruption. It would be more convincing to demonstrate the ability of SCIMPN peptide to block SCIMP antibody binding to exosomes in a flow cytometry based NTF assay. Authors should also verify if SCIMP with truncated N-terminus used in Fig. 4C is still expressed and targeted to cellular membranes/exosomes.

Response:

Thank you for pointing out the defect in proving the SCIMP N-terminus at the outside of exosome by ELISA. According the structure analysis, SCIMP protein has the transmembrane domain, and by the flow cytometry to detect the SCIMP N-terminus, we used the anti-his antibody and as shown in **attached figure 7**, the anti-his antibody could not stain the HEK293 cells which transiently overexpressed SCIMP-his protein, which could well stain these cells penetrated with detergent (Trixton100), which means the “6×his” tag was at the inside of the cell membrane. For lack of penetrating method to exosome, we used the antibody to SCIMP N-terminus, the antibody to SCIMP C-terminus and the antibody to his tag to stained the exosomes purified from the supernatant of HEK293 cells transiently overexpressed SCIMP-6×his, as shown in **attached figure 8a**, only the antibody to SCIMP N-terminus could well stain the exosomes, which indicated on the exosome membrane, the SCIMP N terminus is at outside.

The exosomes purified from the supernatant of SCIMP-6×his overexpressing HEK293 cells were captured by the anti-CD9 capture beads, and then stained by the antibody to SCIMP, CD63, and 6×his. As shown in the **attached figure 8b**, the SCIMP antibody but not the 6×his antibody could well stain the exosomes.

Attached Figure 7. The HEK293 cells transiently transfected with SCIMP-6×his vector or the empty vector, which were fixed with or without membrane penetrated by Triton-100, and the stained by the anti-6×his antibody or the isotype antibody and the fluorescence (FITC) of the cells was detected by flow cytometry.

Attached Figure 8a. The exosomes purified by the ultracentrifuge method from the supernatant of HEK293 transiently transfected with full-length SCIMP vector or N

terminal truncated SCIMP vector, and the exosomes stained with SCIMP antibody (FITC) and CD63 antibody (PE) and detected by nano-particle flow cytometry.

Exosome purified from the supernatant of SCIMP-6Xhis expressing HEK293 by ultracentrifuge method

Attached Figure 8b. By using the FITC-SCIMP antibody and APC-anti-6×his antibody in the SCIMPexo detection kit, we found the C-terminus of SCIMP was mainly at the inside of the exosomes purified from the supernatant of SCIMP-6×his overexpressing HEK293 cells.

5. Potential shortcoming of the data on FPR receptor activation by SCIMP N-terminus is the fact that for experiments demonstrating FPR involvement in Fig.7A-E, SCIMPpro was purified from bacteria and therefore would have formylated N terminus a prototypic FPR ligand. Its preparation could also be contaminated by additional bacterial proteins activating FPRs. In addition, Methods section seems to suggest that SCIMPN peptide was also prepared in bacteria. Authors must clarify if SCIMP N was synthetic peptide or produced in bacteria. Bacterial origin of SCIMPN would severely compromise majority of FPR-related data in the manuscript. Finally, in Fig7A-E, negative control, i.e. cells not expressing FPRs, should be shown. The sequence of SCIMP-N peptide should also be provided.

Response:

Thank you for pointing out this defect in the description of the method. And the detail of the preparation of SCIMP protein and SCIMP N peptides were added and highlighted in the methods. For the SCIMP protein was produced by bacteria, to avoid the chance of the components from bacteria contaminating the SCIMP protein to impact the chemotaxis and others assay, we purified the SCIMP protein with NTA beads and the small sized bacteria components were excluded with ultrafiltration tube (M.W=10 kD)

and washed the protein with sterile PBS for 5 times, and the SCIMP N peptides were chemically synthesized and the purity of the peptides were more than 98%. The chemotaxis activity of the SCIMP protein and SCIMP N peptides were verified by the chemotaxis assay with FPRs overexpressing HEK293 cells. On the other hand, another human sourced protein, previous reported to be a chemokine like protein by our team, PSMP, as a control protein, which was also produced and purified with the same method, and the protein showed no chemotaxis activity to the FPRs overexpressing HEK293 cells in the chemotaxis experiments.

Thank you for the recommendation the results of the negative control, HKE293 cell transiently transduced the empty vector were added in the results and figure. The peptides sequence of the SCIMP was provided in the method section as well. As shown in the attached figure 9, the SCIMP protein purified and expressed from the bacteria could chemoattract the FPR1/2 overexpressing HEK293 cells as well as the chemo-synthesized purified SCIMP N peptides, but not the control protein, PSMP, which also purified and expressed from the bacteria with the same method. Based on this observation, the contamination of bacteria components in the purified protein that could influence the chemotaxis activity could be excluded. For the calcium flux assay, we also tested whether the purified SCIMP protein or fMLF could induce the calcium flux in HEK293 transfected with empty vector or not, and no calcium flux signal was observed in attached figure 10. For the directly binding assay in vitro, the detail of experiment process and more discussion of related results were added in the manuscript, and the cell membrane debris of HEK293 cells transfected with the empty vector (pcDNA3.1) were used as the control samples in these experiments.

Attached Figure 9

Attached Figure 9. The chemotaxis activity of the control protein (prokaryotic PSMP, expressed and purified from bacteria), SCIMP protein (expressed and purified from bacteria) and SCIMP N terminus peptides (biosynthesized and purified by HPLC) to FPR1 or FPR2 overexpressing HEK293 cells in chemotaxis assay (Boyden chamber, chemoattracting time was 9 hours) with the concentration of 1, and 10 nM.

Attached Figure 10

Attached Figure 10. The calcium flux of the HEK293 cell transfected with the empty vector was detected by a time coursed confocal, and at the 20 second and 180 second of the observation the 200 nM SCIMP protein and 100 nM fMLF were added into the supernatant.

6. Experiments in Fig. 7G should be explained more clearly, especially arrows in the left panel and Kd calculation method.

Response:

Thank you to point out the defect, the details including the experiment process and results description was added and highlight in the revised manuscript.

7. It is important to show that endogenous levels of SCIMP in endosomes are able to attract neutrophils to lungs. This authors attempted to do in Fig. 8D, G. However, there they used exosomes from E.coli treated Raw cells which could be contaminated with bacteria. Using the exosomes from SCIMP^{-/-} and WT bone marrow derived macrophages would provide necessary controls missing in the current Fig. 8D, G.

Response:

Thank you for pointing out the question, and it's important to elucidate the endogenous SCIMP positive exosome in the lung to chemoattract neutrophils. To data this, we performed two experiments, one experiment was to perfuse the BAL of the wild type mice or SCIMP knocked out mice with ALI, as shown in **attached figure 11a**, exosome counts in the BALF of two groups had no significant difference, while the percentage of SCIMP positive exosome from the SCIMP KO mice were much lower than those from the wild type mice. The other experiments was the chemotaxis activity of the exosomes secreted by the bone marrow derived macrophages from the wild type mice and SCIMP KO mice, as shown in **attached figure 11b**, after the exosome perfused into lung for 4 hours, the neutrophils chemoattracted into the lung were detected by flow cytometry, the exosome from the BMDM of the SCIMP KO mice showed a deficiency in neutrophils chemotaxis, which supported the assumption of SCIMP existing on the exosome in lung was important for the neutrophils chemotaxis, especially for the FPRs positive neutrophils.

Attached Figure 11a

Attached Figure 11a. The exosomes from the supernatant of wild type or Scimp knocked out mice sourced bone marrow derived macrophages (co-cultured with L929 cells) were purified and stained by CD63 antibody (PE) and SCIMP antibody (FITC), and the samples were detected by nano-particle flow cytometry, finally the count of CD63 positive exosomes in two groups and the percentage of SCIMP positive particles in the CD63 positive exosomes were calculated (WT versus SCIMP KO = 5 versus 5).

Attached Figure 11b

Attached Figure 11b. The exosomes from the supernatant of wild type or SCIMP knocked out mice sourced BMDM (incubated with L929 cells) were collected and purified, and the same amount of exosomes were perfused into the lung of wild type C57 mice, and four hours later, the BALF from the two groups were collected, finally the percentage of neutrophils (CD11b+Ly6G+) and FPRs+ neutrophils (CD11b+Ly6G+FPRs+) compared to the alveolar macrophages (CD11b+CD11c+) were analyzed (the count of the individuals in each group n =5) .

8. Potential weakness of this manuscript is the design of ALI mouse model, where massive E.coli infection leads to death within hours. Here, higher neutrophil infiltration is useful to combat the high bacterial load. It is good evidence of the ability of SCIMP exosomes to attract neutrophils and have impact in vivo. On the other hand, it is unclear how exactly this situation is relevant for human patients, where the infections develop with slower kinetics and more neutrophils could also mean more tissue damage. These issues should be properly discussed/explained.

Response:

Thank you for pointing out this defect in our mechanism. According to the previously

reports, the neutrophils chemoattracted to the lung when pneumonia had two effects, one is to help to clean the pathogens and the other is to release harmful inflammatory signals to ameliorate the injury. And lots of studies tried to elucidate these to processes by establishing the LPS induced ALI model, which could exclude the proliferation and the expansion of the live bacteria in vivo leading to the death. In our study of LPS induced ALI model, to perfuse the lung with LPS and SCIMP N peptides could not elevate the survival rate (**attached figure 12**). In clinical, with the help of the antibiotic chemicals, the patients, especially with a compromised immune system, was infected with bacteria could survival for a long term and a high rate, and the neutrophils were treated as the inflammatory factor. However, at the early stage of the infection, if the patients pretreated with the SCIMP components, the invading bacteria might be eliminated timely and the subsequent classic cytokines and neutrophils would not increase so high that could damage the tissue. For the middle or late stage of the infection, the antibiotic chemicals or the neutrophils infusion have been proven more efficiency (doi: 10.1126/scitranslmed.abb1069). Moreover, we also added this discussion into the discussion sections.

Attached Figure 12

Attached Figure 12. The ALI model was established by bronchial perfusion of LPS, and the survival rate of the models with the bronchial perfusion of SCIMP N peptides (MF8) or PBS was analyzed (n = 6).

9. SCIMP N-terminus can be expected to be exposed on all SCIMP expressing cells. Is cell-expressed SCIMP accessible to antibody staining? Could it also serve as neutrophil chemoattractant?

Response:

Thank you for pointing out this question. Cells expressing chemoattracting components is not an ideal method to chemoattract the targeting cells. In this study, we found the SCIMP also expressed on the cell membrane and the N-terminus was outside. If we treated the cells or mice with the exosome secreting inhibitor GW4869, the chemotaxis effect would become weak to the neutrophils. Including the commercial antibody of SCIMP or our antibody of SCIMP both could stain the SCIMP exogenously and endogenously expressing cells (**attached figure 13**).

Attached Figure 13. The HEK293 cells expressing SCIMP-EGFP recombinant protein were established and the distribution of SCIMP-EGFP recombinant protein in the cells were observed under the fluorescence microscope (upper-left, bar = 20 μm); The HEK293 cells transfected with SCIMP-6×his vector or empty vector (pcDNA3.1) were stained by the SCIMP antibody and observed under the confocal (upper-right, bar = 20 μm) and cytometry (lower).

10. Human SCIMP has two alternative translation starts generating two different N termini. Do they both have chemotactic properties? How this affects interpretation of the data?

Response:

Thank you for pointing out these questions. As shown in **attached figure 14**, the

SCIMP N-terminus has two translation start points, so we obtain the SCIMP protein contained fraction from exosome of SCIMP overexpressing HEK293 cells on the sds-page stained with Coomassie Brilliant Blue and sequence the N terminus of protein. And according to the result (**attached figure 14**), the main type of SCIMP protein in the exosome was the first one.

Attached Figure 14

Attached Figure 14. The amino acids in the mammalian expressed and purified SCIMP were measured by mass spectrometry and the first five amino acids in the sample were shown and labelled (red arrow).

Reviewer #2 (Remarks to the Author):

In the manuscript entitled “Exosomal secreted SCIMP regulates communication between macrophages and neutrophils in pneumonia”, Pei and colleagues present their novel findings on exosomal SCIMP from macrophages after bacterial infection. Functionally, exosomal SCIMP has chemotactic activity and activates neutrophils through FPR1/2. This is a carefully done study with both gain-of-function and loss-of-function approaches. The findings are novel and interesting. However, enthusiasm is dampened by several flaws in the study design and experimental methods.

Major comments:

1. Is SCIMP a secreted protein? As shown in Fig. 1, chemotaxis activity was observed in SCIMP-containing supernatant from SCIMP over-expressing HEK293 cells. Unfortunately, the authors failed to determine if soluble SCIMP, exosomal SCIMP, or both of them contributed to the chemotactic effect. Therefore, the soluble and exosomal SCIMP needs to be quantified using ELISA. In addition, the chemotaxis activity should be carefully compared throughout the work given that soluble SCIMP may play a major role in both in vivo and in vitro conditions.

Response:

Thank you for pointing out this question. In the experiment of SCIMP transiently overexpressed by HEK293 cells, as shown in **attach figure 15**, in the exosome free supernatant the SCIMP could hardly be detected by the SCIMP antibody in Western blot, while in the exosome from the supernatant purified by the ultracentrifuge method the SCIMP protein could be strongly stained with SCIMP antibody as well as the known exosome markers. However, we tried to accumulated the soluble SCIMP protein in the 500 mL supernatant of SCIMP-His transiently overexpressing HEK293 cells by the NTA beads, and about 0.5 ug SCIMP-his protein (highly contaminated with other protein) could be enriched (**attached figure 16**), which indicated SCIMP protein might be secreted by a soluble pattern, but mainly by an exosomal pattern. Given these findings, we tested the chemotaxis activity of exosomal SCIMP, soluble prokaryotic

SCIMP protein and SCIMP N terminal peptides, and all of them had chemotaxis activity to peripheral neutrophils (**attached figure 17**).

Attached Figure 16

Attached Figure 16. The SCIMP-6×his protein in the supernatant and cell lysate expressed by HEK293 cells and the SCIMP protein enriched from supernatant by NTA beads was detected by anti-6×his antibody in the Western blot; the enriched protein were analyzed by the SDS-PAGE and stained by Coomassie blue.

Attached Figure 17

Attached Figure 17. The chemotaxis activity of SCIMP contained exosome, purified SCIMP protein, and SCIMP N peptides to peripheral neutrophils (PBN: peripheral neutrophils, PBM: peripheral monocytes, PBL: peripheral lymphocytes) were analyzed in the TaxiScan system and Boyden chamber chemotaxis assay.

2. The exosomes are not well characterized. The authors briefly characterized HEK293-derived exosomes. In contrast, exosomes from animals or human subjects were not validated.

Response:

We thank you for pointing put this question. We have obtained the SCIMP enriched exosome from the supernatant of SCIMP transiently overexpressing HEK293 cells were stained by the SCIMP antibody in Western blot, as well as the antibodies of CD9, CD63, CD81, TSG101, which proved the SCIMP was mainly secreted by an exosomal way (**attached figure 18**). According to our previously study, the endogenous SCIMP protein in the body fluid or serum from the human or animal without infection could hardly detected by Elisa or Western blot. Given these, we tried to detect the SCIMP on exosomes by nano particle flow cytometry, in which the exosomes were stained by the SCIMP antibody and CD63 antibody. As a result, After the exosomes in the BALF of ALI model were purified by SEC column, the SCIMP attendance on the exosomes could be verified (**attached figure 19**)

Attached Figure 18. The SCIMP protein and exosome markers (CD9, CD63, CD81, TSG101) from the cell lysate, exosome purified from the supernatant (by the ultracentrifuge method), and the exosome free supernatant of HEK293 cells transiently expressing SCIMP-6×his protein were stained in the Western blot.

Attached Figure 19

Attached Figure 19. The exosomes from BALF of the ALI model were collected and purified by the SEC column, and validated by staining the exosome samples with the CD63 antibody (PE) and SCIMP antibody (FITC) and through nano-particle flow cytometry.

3. It is very interesting that SCIMP protein was secreted via exosome from the RAW264.7 cells after stimulation with denatured E.coli but not after stimulation with LPS. Did LPS or bacterial infection induce SCIMP expression at the mRNA level? A previous study has reported that SCIMP promotes selective proinflammatory cytokine responses by direct modulation of TLR4, which is a well-known receptor involved in LPS recognition and signal initiation. In this scenario, it would be better to further investigate the mechanism, although the authors briefly stated in the discussion.

Response:

Thank you for pointing out this question. We performed the experiment of the BMDM (bone marrow derived macrophages) stimulated with the heat denatured bacteria or LPS, then we detected the SCIMP mRNA level at different time points, and we found the at 12 hours post the stimulations, the mRNA level in the macrophages with the stimulation of bacteria and LPS increased (**attached figure 20**), which indicated that SCIMP might play a role in the ALI model from the early to late stages. Additionally, as shown in **attached figure 21**, the perfusion of SCIMP N peptides could not elevate the survival rate of ALI model induced with LPS, which implied the different mechanisms of SCIMP participating in the pulmonary inflammation induced with LPS or bacteria.

Attached Figure 20

Attached Figure 20. The BMDM from the wild type C57 mice were induced by L929 for 48 hours, and sorted by anti-F4/80 magnetic beads, and then the cells were incubated with PBS, heat denatured *S.aureus* (MOI = 100:1), heat denatured *E.coli* (MOI = 100:1), or LPS (100nM) respectively, at 6 hours, 12 hours and 24 hours after the incubation the mRNA from the cells were extracted and the SCIMP transcriptional level were measured by Q-PCR.

Attached Figure 21. The ALI model was established by bronchial perfusion of LPS, and the survival rate of the models with the bronchial perfusion or abdominal injection of SCIMP N peptides (MF8) or PBS was analyzed (n = 6).

4. The methods and figure legends are not informative. Please describe the details of each method, especially for in vivo experiments. For instance, how the mice were anesthetized? What was used for perfusion? How many mice were used for each experiment?

Response:

Thank you for pointing out this defect, we had added and highlighted the detail description of the figure legends and methods in the manuscript.

5. Fig. 5 and 9 indicated that soluble SCIMP (SCIMP peptides) alone has similar effects as exosomal SCIMP. If so, why not focus on soluble SCIMP rather than exosomes? It is not clearly stated.

Response:

Thank you for pointing out this question. In our previous study, even though the full length SCIMP protein and SCIMP N terminal peptides were found to have chemotaxis activity, we found the SCIMP protein from the cell supernatant, body fluid or serum was mainly attendant on the exosomes. It might be because the SCIMP protein is necessary in the exosome secretion in macrophages, lack of certain enzyme to digest the N terminal peptides of SCIMP into soluble pattern or secreting signaling peptides, or the targeting function of the exosomes depending on SCIMP, soluble SCIMP could hardly be detected. And we also found SCIMP protein, especially the SCIMP N terminal peptides, kept the chemotaxis activity. Even though exosomal SCIMP could imitate the biological processes in the ALI model more precisely, there had substantial difference between the exosomes secreted by primary alveolar macrophages and cell lines, including the exosomes encapsulated RNAs, cytokines and other components. Therefore, we tried to elucidate the SCIMP chemotaxis ability and its role in ALI by mainly using SCIMP N terminal peptides, which also could be developed as the anti-pneumonia reagents.

Minor comments:

1. Please show the dot plot to indicate the sample size.

Response:

Thank you for pointing out this question, the statistical data represented by column has been changed into dot plot format in the manuscript as well as the replicated count.

2. What does “the colocalization of SCIMP and LAMP2A/EEA1 was weak” mean in

Figure S1?

Response:

Thank you for pointing out this question. We used the colocalization calculator from “ImageJ” to quantize the colocalization of SCIMP to LAMP2A or EEA1, as shown in **attached figure 22**, at 4 hours post incubation, the reduced colocalization rate (Pearson’s R value) of SCIMP to LAMP2A in the cells incubated with bacteria (from 0.67 to 0.62) is similar to that in the cells incubated with LPS (from 0.95 to 0.95), while the reduced colocalization rate of SCIMP to EEA1 in the cells incubated with bacteria (from 0.61 to 0.44) is higher than that in the cells incubated with LPS (from 0.84 to 0.82). The reduced colocalization of SCIMP to EEA1 with the incubation of bacteria might indicate the SCIMP translocating from the early endosome to late endosome during the exosome generation processes. And we have modified the description of the figure legend to avoid the confusion.

Attached Figure 22

Attached Figure 22. The colocalization of exogenously expressed SCIMP-GFP to LAMP2A or EEA1 in the RAW264.7 cells with the stimulation of heat denature E.coli (MOI = 100) or LPS (1 μM) at different time points (0, 1, 4 hours); the colocalizing score was calculated by the “coloc2” algorithm and evaluated by the Pearson’s score.

3. Several methods were not described, e.g. cell culture, transfection, ELISA, etc.

Response:

Thank you for pointing out this defect. We have added and highlighted the description of these methods into the manuscript.

REVIEWER COMMENTS

Reviewer #1 (Remarks to the Author):

The authors addressed majority of my concerns. However, some issues still remain unresolved or have arisen after addition of the new results:

1. The data showing that SCIMP antibody reacts with SCIMP overexpressed in HEK cells or bacteria are now convincing. With respect to this issue, I only suggest a few clarifications and changes in the way how the data are presented or discussed. In Fig. S15, the right panel shows that the bacterially produced SCIMP has apparent MW of 15 kDa. However, the area around 15 kDa on the coomassie-stained gel with the sample of purified SCIMP does not contain any strong bands, suggesting that SCIMP purification was rather inefficient. If this is the case, authors should be explicit about it in the manuscript text. In addition, Fig. S14 implies that SCIMP purification procedure included cutting out the area of the gel corresponding to SCIMP followed by SCIMP extraction. This is not a commonly used standard procedure and should be described in more detail in the Methods section or in the supplement. Authors should also clearly indicate which area of the gel was cut. There is also a discrepancy between the apparent MW of SCIMP produced in bacteria (15 kDa) and in HEK cells (20 kDa). Was full-length SCIMP expressed in bacteria? This information has to be provided (and if not full length, then exactly which part of SCIMP sequence was used to generate the antibody). Explanation or comments on differential mobility of SCIMP produced in bacteria and HEK cells should also be included. It is also important to present the data comparing SCIMP- and empty vector-transfected HEK from “attached fig. 4” with appropriate loading control (e.g. CD9) directly in the article or its supplement. A loading control is also missing in the right panel of Fig. S15 (total protein staining would be sufficient here).

Regarding the endogenous SCIMP detection in Raw cells treated with E.coli in Fig. 1F (note the full size blot in the powerpoint dataset file and also my previous evaluation), the problem with antibody staining the entire lanes has not been addressed. With endogenous protein, signal to background ratio may be smaller and any staining of bacterial proteins may become more prominent. To address this issue, authors can perform the same experiment with BMDMs from WT and SCIMP^{-/-} mice to reliably distinguish the bands representing SCIMP. They also have to add an appropriate loading control.

Since large part of the data presented in this manuscript are dependent on this antibody, it is critical to convince the reader that the antibody is functional and sufficiently specific.

2. Axes labeling is missing in Fig. 1D. In addition, the data from non-transfected controls as shown in “attached figure 5” have to be presented in the main figure together with the plot from transfected cells. It is key in helping the reader judge the specificity and functionality of this method. In the “attached figure 5” the gates in the left and right plots are not placed in the same position, artificially increasing the percentage of SCIMP-positive particles in SCIMP-transfected cells. What is the percentage after this is corrected? Presentation of the data could be enhanced by gating on all CD63⁺ particles and from this

gate creating an overlay of SCIMP-FITC histograms from SCIMP-transfected and non-transfected cells and adding this to the Fig. 1D.

3. The problems with CD81 and CD63 detection on the Western blot in Fig 1C has not improved since the previous version. The antibodies do not appear to be functional (at least on Western blot) and the data at present form are not usable. However, the use of additional markers CD9 and TSG101 helped circumvent this issue. Some tetraspanin-specific antibodies are known to work better on non-reduced samples. Authors can try running non-reduced samples or provide a verification other than just a molecular weight. Otherwise, they should remove the CD81 and CD63 blots from the manuscript. References describing all the above mentioned proteins as exosome markers should be placed in the manuscript text.

4. Quantification of SCIMP co-localization with endosomal markers in Fig. S1C lacks statistical evaluation. Are the observed differences significant? Also, why are the co-localizations in non-stimulated cells so different between E.coli and LPS (0 hrs) samples?

5. Is SCIMPpro prepared in the same way as the material shown in Fig. S15 left panel? Majority of the proteins there seem to be contaminating bacterial proteins (see above). Authors should provide the data on the purity of SCIMPpro (coomassie-stained gel).

6. The exact sequence of SCIMPN peptide has to be shown in the manuscript. Authors should also comment on the fact that (as far as I could find out) mice only have the short version of SCIMP N-terminus. Thus murine SCIMP likely only has 7 extracellular amino acids (as opposed to human 18) Authors should discuss this issue and how it affects the interpretation of their data. Authors also write about murine SCIMPN on multiple occasions. Does it really contain murine sequence?

7. Spelling and grammar throughout the entire manuscript has to be corrected.

Reviewer #2 (Remarks to the Author):

The authors addressed most of the questions by including more data and details in Methods. The quality of the manuscript has been improved. Minor concerns are listed as follows:

1. There are numerous typos and grammatical errors in the manuscript, which will need an extensive English language edit.

For example,

Page 2 line 43 “absent > absence”.

Page 7 line 157 “that from that from > that of”

Page 9 line 189 “were > was ”

Page 10 line 207 “no > not”

Page 10 line 210 “when > which”

Page 15 line 319 “obvious > obviously”

Page 17 line 350 “desensitized > desensitize ”

Page 19 line 390 “alternated > be altered”

Page 23 line 438 “recognizing > recognize”

Page 23 line 452 “phonotype > phenotype”

Page 25 line 496 “described > been described”

Page 28 line 560 “extracted > extract”

Page 28 line 581 “Ang The the cells > The cells”

This list is not exhaustive. Please check throughout the manuscript and make amendments

2. “The mice were used for experiments at 6–8 weeks of age and the gender was male or female”. The statement is confusing. Please specify the sex of the mice used in the experiments.

Point-by-point response to the reviewers' comments:

Reviewer #1 (Remarks to the Author):

The authors addressed majority of my concerns. However, some issues remain unresolved or have arisen after addition of the new results:

1. The data showing that SCIMP antibody reacts with SCIMP overexpressed in HEK cells or bacteria are now convincing. With respect to this issue, I only suggest a few clarifications and changes in the way how the data are presented or discussed. In Fig. S15, the right panel shows that the bacterially produced SCIMP has apparent MW of 15 kDa. However, the area around 15 kDa on the Coomassie-stained gel with the sample of purified SCIMP does not contain any strong bands, suggesting that SCIMP purification was rather inefficient. If this is the case, authors should be explicit about it in the manuscript text. In addition, Fig. S14 implies that SCIMP purification procedure included cutting out the area of the gel corresponding to SCIMP followed by SCIMP extraction. This is not a commonly used standard procedure and should be described in more detail in the Methods section or in the supplement. Authors should also clearly indicate which area of the gel was cut.

There is also a discrepancy between the apparent MW of SCIMP produced in bacteria (15 kDa) and in HEK cells (20 kDa). Was full-length SCIMP expressed in bacteria?

This information has to be provided (and if not full length, then exactly which part of SCIMP sequence was used to generate the antibody). Explanation or comments on differential mobility of SCIMP produced in bacteria and HEK cells should also be included. It is also important to present the data comparing SCIMP- and empty vector-transfected HEK from “attached fig. 4” with appropriate loading control (e.g. CD9) directly in the article or its supplement. A loading control is also missing in the right panel of Fig. S15 (total protein staining would be sufficient here). Regarding the endogenous SCIMP detection in Raw cells treated with E.coli in Fig. 1F (note the full size blot in the powerpoint dataset file and also my previous evaluation), the problem with antibody staining the entire lanes has not been addressed. With endogenous protein, signal to background ratio may be smaller and any staining of bacterial proteins may become more prominent. To address this issue, authors can perform the same

experiment with BMDMs from WT and SCIMP^{-/-} mice to reliably distinguish the bands representing SCIMP. The SCIMP band on BMDM cell lysate, they also must add an appropriate loading control. Since large part of the data presented in this manuscript are dependent on this antibody, it is critical to convince the reader that the antibody is functional and sufficiently specific.

Response:

Thank you for providing these constructive suggestions. For the SCIMP antibody used in this study was made in our laboratory, it is important and necessary to provide enough details of the usage of the antibody in the multiple experiments and the evaluation of the specificity and sensitivity in the supplementary materials. The protein to immunized the rabbit is the SCIMP full length protein fused with 6Xhis tag expressed in the bacteria, BL21DE3. After the SCIMP-6Xhis protein were accumulated by the NTA column, the protein samples were loaded on the SDS-PAGE. In Fig.S15, the lanes of the eluted fractions did not show the final product of the purified SCIMP protein, because we found it was not easy to obtain the SCIMP protein purified from the bacteria lysate might due to the specific structure of SCIMP protein. We found the salt concentration (mainly was sodium chloride) could affect the NTA beads specifically binding to the expressed His-tagged protein in the bacteria lysate, and as shown in **Attached Figure 1** the washing buffer with a higher salt concentration was helpful to get a relative high purity of SCIMP protein that was sufficient for antibody immunization with the targeting band gel-cutting.

And after we optimized the salt concentration and imidazole concentration in the washing buffer and the elution buffer, as shown in Fig.S15a and Attached Figure 2, the purification of the obtained SCIMP protein was acceptable for the subsequent chemotaxis assay in this study.

Attached Figure 2

As for the molecular weight of SCIMP in the Fig.S15 and other related results, the calculated M.W of full length SCIMP is 16.6 KD, and in the Western blot result and the SDS-PAGE result in Fig.S15 and related figures the SCIMP protein band was located around the 15 KD marker, and in the Attached Figure 3, the SCIMP expressed by HEK293 was located around the 25 KD marker, which is larger than the prokaryotic expressed SCIMP protein.

Attached Figure 3

The SCIMP protein expressed by bacteria and HEK293 cells was full length protein, and more details of the related results in the manuscript have been added to avoid the confusion. The explanation on differential mobility of SCIMP produced in bacteria and HEK293 cells had been added into the discussion section.

As shown in the **Attached Figure 4**, the exosomal control, CD9, in the samples of HEK293 transfected with empty vector or SCIMP vector were stained as well as the SCIMP protein, and the SCIMP overexpression in HEK293 cells does not affect the exosome producing which was confirmed by the related experiments in this study.

Attached Figure 4

The total protein of the bacteria lysate stained with SCIMP antibody has been added into the Fig.S15.

In experiment about Fig.1F, the RAW cells were incubated with PBS, LPS or heat-killed *E.coli* (121°C for 20 min) for 30 minutes, 1 hour and 2 hours. The exosomes in the serum-free supernatant were extracted and the endogenous exosomal SCIMP protein was stained by the SCIMP antibody in Western blot. We found that only the heat-killed *E.coli* could induced the RAW cell to release the SCIMP abundant exosomes, which was also observed in the nanoparticle flow cytometry assay. In the whole blot image of Fig.1F, the bands located around 25 kD was specific, which should be the SCIMP protein, and there were also bands located around 45 KD which should not be the SCIMP protein, for we never observed the SCIMP locating at around 45 KD in other

experiments, which might be due to the primary or secondary antibody recognizing the proteins contaminated from the bacteria component.

Compared with the SCIMP overexpressing cell lysate samples, the endogenous SCIMP level in the primary macrophages is lower, after the bone marrow cells from the wild type mice and *SCIMP* knocked out mice and the BMDMs were induced, the endogenous SCIMP protein were stained with SCIMP antibody and the control protein CD19 (Attached Figure 5). We totally agree the importance of the specificity of an antibody to the reliability of the result, beside to provide more evidence to prove the reliability of the rabbit hosted poly clonal antibody of SCIMP, we also developed the monoclonal antibody of SCIMP from mouse and employed in the future projected.

Attached Figure 5

2. Axes labeling is missing in Fig. 1D. In addition, the data from non-transfected controls as shown in “attached figure 5” have to be presented in the main figure together with the plot from transfected cells. It is key in helping the reader judge the specificity and functionality of this method. In the “attached figure 5” the gates in the left and right plots are not placed in the same position, artificially increasing the percentage of SCIMP-positive particles in SCIMP-transfected cells. What is the percentage after this is corrected? Presentation of the data could be enhanced by gating on all CD63+ particles and from this gate creating an overlay of SCIMP-FITC histograms from SCIMP-transfected and non-transfected cells and adding this to the Fig. 1D.

Response:

Thank you for pointing out this defect. And the left threshold of the gate “P3” was

different, might due to we used the “CytExpert” to analysis the data and forgot the applied the setting to all tubes, sorry about the confusion we made. The new gating strategy were shown in **Attached Figure 6** and added into the **Fig.1D**, the related statistic data was reanalyzed as well. In detail, in the exosome samples purified from the supernatant of the HEK293 over expressing SCIMP, the percentage of SCIMP positive particles in the CD63+ particles are more than 75%, while the percentage of those from the HEK293 transfected with empty vector is less than 25%.

3. The problems with CD81 and CD63 detection on the Western blot in Fig 1C has not improved since the previous version. The antibodies do not appear to be functional (at least on Western blot) and the data at present form are not usable. However, the use of additional markers CD9 and TSG101 helped circumvent this issue. Some tetraspanin-specific antibodies are known to work better on non-reduced samples. Authors can try running non-reduced samples or provide a verification other than just a molecular weight. Otherwise, they should remove the CD81 and CD63 blots from the manuscript. References describing all the above mentioned proteins as exosome markers should be placed in the manuscript text.

Response:

Thank you for your suggestions. And we tried to prepare and run the non-reduced samples on the non-denaturing PAGE and stained the CD81, CD63 with new antibodies purchased from Abcam, and the stained bands on the membrane was shown in **Attached Figure 7**, as well as in the **Fig 1C**. The unspecific bands stained with CD81 antibody were still exist, and the bands located at 20-25 KD was stronger than the previous, the band stained with CD63 antibody that located at 40-55 KD was much specific. Additionally, the references describing all the above-mentioned proteins as exosome markers has been added in the manuscript text.

Attached Figure 7

4. Quantification of SCIMP co-localization with endosomal markers in Fig. S1C lacks statistical evaluation. Are the observed differences significant? Also, why are the co-localizations in non-stimulated cells so different between E.coli and LPS (0 hrs) samples?

Response:

Thank you for pointing out this defect. To obtain the statistical evaluation of the co-localization, at least ten cells in one group were used to quantify the co-localization of SCIMP and endosomal markers. The previous quantification data was from a part of one cell, which might due to the cell individual heterogeneity, the co-localization score was higher in the LPS treated group than in the bacteria treated group. To avoid the bias, we calculated at least ten cells in each time point group. As shown in **Fig.S1 and Attached Figure 8**, after stimulated with bacteria, not only the SCIMP amount, but also the colocalization of SCIMP to endosomal markers were decreased with a significant difference with that treated with LPS, which might mean the SCIMP was

secreted along with the exosomes, and coordinating to the observation of the SCIMP positive exosomes increased in the supernatant of macrophages with the stimulation of bacteria.

Attached Figure 8

5. Is SCIMPpro prepared in the same way as the material shown in Fig. S15 left panel? Majority of the proteins there seem to be contaminating bacterial proteins (see above). Authors should provide the data on the purity of SCIMPpro (coomassie-stained gel).

Response:

Thank you for pointing out this question. We carefully checked the results and experiment log, and found the previous Fig.S15 was not the final product. In Fig.S15, the lanes of the eluted fractions did not show the final product of the purified SCIMP protein, for we found it was not easy to obtain the SCIMP protein purified from the bacteria lysate might due to the specific structure of SCIMP. And after we optimized the salt concentration and imidazole concentration in the washing buffer and the elution buffer, as shown in Fig.S15b and Attached Figure 9, the purification of the obtained SCIMP protein was acceptable for the subsequent chemotaxis assay in this study.

Attached Figure 9

6. The exact sequence of SCIMPN peptide has to be shown in the manuscript. Authors

should also comment on the fact that (as far as I could find out) mice only have the short version of SCIMP N-terminus. Thus murine SCIMP likely only has 7 extracellular amino acids (as opposed to human 18) Authors should discuss this issue and how it affects the interpretation of their data. Authors also write about murine SCIMPN on multiple occasions. Does it really contain murine sequence?

Response:

Thank you for pointing out this defect. The sequences of human and murine SCIMP N terminus peptides were added and highlighted in the manuscript, as well as the discussion of the difference between the human and murine SCIMP N sequence. In this study, the SCIMP N terminus used in all the murine related experiments was murine SCIMP N terminus peptides. To avoid the potent confusion, we also optimized and highlight the related description in the manuscript.

7. Spelling and grammar throughout the entire manuscript has to be corrected.

Response:

Thank you for pointing out this defect. The spelling and grammar throughout the entire manuscript has been corrected by a scientific native English speaker.

Reviewer #2 (Remarks to the Author):

The authors addressed most of the questions by including more data and details in Methods. The quality of the manuscript has been improved. Minor concerns are listed as follows:

1. There are numerous typos and grammatical errors in the manuscript, which will need an extensive English language edit.

For example,

Page 2 line 43 “absent > absence”.

Page 7 line 157 “that from that from > that of”

Page 9 line 189 “were > was ”

Page 10 line 207 “no > not”

Page 10 line 210 “when > which”

Page 15 line 319 “obvious > obviously”

Page 17 line 350 “desensitized > desensitize ”

Page 19 line 390 “alternated > be altered”

Page 23 line 438 “recognizing > recognize”

Page 23 line 452 “phonotype > phenotype”

Page 25 line 496 “described > been described”

Page 28 line 560 “extracted > extract”

Page 28 line 581 “Ang The the cells > The cells”

This list is not exhaustive. Please check throughout the manuscript and make amendments

Response:

Thank you for pointing out this defect. The spelling and grammar throughout the entire manuscript has been corrected by a scientific native English speaker.

2. “The mice were used for experiments at 6–8 weeks of age and the gender was male or female”. The statement is confusing. Please specify the sex of the mice used in the experiments.

Response:

Thank you for pointing out this defect. The gender description of the mice used in this study has been corrected. In brief, in the experiments about ALI models, the mice gender was male.

REVIEWER COMMENTS

Reviewer #1 (Remarks to the Author):

Authors now provide sufficient evidence that their anti-SCIMP antibody is specific and recognizes overexpressed as well as endogenous SCIMP and that it is suitable for use in experiments not including bacteria. However, there are still some outstanding issues with antibody specificity in bacteria-treated samples in Fig 1F, G and H. The specificity of the antibody cannot be determined only by molecular weight matching. The antibody in bacteria-treated samples stained strongly additional proteins (that could be of bacterial origin) and if one of the non-specifically stained bands happens to have similar molecular weight, it can be easily mistaken for SCIMP. Moreover, the double-band considered to be SCIMP in Fig. 1F has lower molecular weight than 25kDa SCIMP shown in other experiments. Finally, the results do not match the observation in Fig. S1, showing that during the first 2 hrs of E-coli treatment (which is the timescale of the experiment in Fig 1F) SCIMP-GFP co-localization with EEA1 and LAMP2A does not change. The experiment in Fig 1F has to be performed with proper controls, which in this case would be SCIMP-negative/downregulated cells as I suggested in my previous reviews. This has to be taken seriously, because the result also affects the interpretation of Figures 1G and 1H.

For Figure 1C authors obtained new antibodies to CD63 and CD81. However, the patterns of bands these antibodies stain in Attached Figure 7 appear remarkably similar (compare left and right panel). The band that authors claim to be CD81 seems to be unique and could represent CD81 even though it is much weaker than other bands on the same membrane. However the CD63-stained band appears also in CD81 staining and is, thus, very likely non-specific.

I also suggest replacing Figure 1D with the entire Attached figure 6 (i.e. including non-transfected samples), because the present figure without negative controls is not sufficient to fully understand the gating strategy.

There still remain grammatical errors in the manuscript, mainly in the newly added text and in the supplementary data.

All the other issues have been resolved.

I will now address the new questions raised by the reviewer.

Comments:

Authors now provide sufficient evidence that their anti-SCIMP antibody is specific and recognizes overexpressed as well as endogenous SCIMP and that it is suitable for use in experiments not including bacteria.

However, there are still some outstanding issues with antibody specificity in bacteria-treated samples in Fig 1F, G and H. The specificity of the antibody cannot be determined only by molecular weight matching. The antibody in bacteria-treated samples stained strongly additional proteins (that could be of bacterial origin) and if one of the non-specifically stained bands happens to have similar molecular weight, it can be easily mistaken for SCIMP. Moreover, the double-band considered to be SCIMP in Fig. 1F has lower molecular weight than 25kDa SCIMP shown in other experiments. Finally, the results do not match the observation in Fig. S1, showing that during the first 2 hrs of E-coli treatment (which is the timescale of the experiment in Fig 1F) SCIMP-GFP co-localization with EEA1 and LAMP2A does not change.

The experiment in Fig 1F has to be performed with proper controls, which in this case would be SCIMP-negative/downregulated cells as I suggested in my previous reviews. This has to be taken seriously, because the result also affects the interpretation of Figures 1G and 1H.

For Figure 1C authors obtained new antibodies to CD63 and CD81. However, the patterns of bands these antibodies stain in Attached Figure 7 appear remarkably similar (compare left and right panel). The band that authors claim to be CD81 seems to be unique and could represent CD81 even though it is much weaker than other bands on the same membrane. However, the CD63-stained band appears also in CD81 staining and is, thus, very likely non-specific.

I also suggest replacing Figure 1D with the entire Attached figure 6 (i.e. including non-transfected samples), because the present figure without negative controls is not sufficient to fully understand the gating strategy.

There remain grammatical errors in the manuscript, mainly in the newly added text and in the supplementary data.

All the other issues have been resolved.

Response:

1. In the experiment involving RAW264.7 secretion of SCIMP-positive exosomes stimulated by heat-inactivated bacteria, we optimized the following experimental conditions:

- Reduced the co-incubation ratio of bacteria and cells from the original 1000:1 to 100:1 to minimize the impact of bacterial components on western blot experiments.
- Added centrifugation steps to better separate bacterial components when collecting the supernatant.
- Designed shRNA for the *Scimp* gene, introduced shRNA into RAW cells using lentivirus to obtain stable *Scimp* knockdown cell lines.
- Optimized antibody incubation time, reducing the primary antibody incubation time to 1 hour at room temperature.
- Used a more specific secondary antibody.

As shown in Figure 1F (and the corresponding figure), in this experiment, SCIMP protein in exosomes was well stained, reducing interference from previous nonspecific bands. Additionally, following the reviewer's suggestion, we found a very weak staining band in exosomes from the *Scimp* knockdown cell line at 6 hours, further confirming the specificity of the bands shown in the western blot. Regarding the time points of SCIMP secretion, SCIMP staining bands appeared 2 hours after stimulation and significantly increased at 6 hours, consistent with the co-localization and secretion process we observed in the confocal experiments.

2. In the experiment proving that the overexpression of SCIMP is mainly through exosome secretion, we reordered new CD81 and CD63 antibodies from *Abcam*, and restained the original samples. As shown in Figure 1C and the

corresponding supplementary figure, western blotting could effectively stain specific bands. We have replaced the corresponding bands accordingly.

3. Thanks to the reviewer for the suggestion regarding the gating strategy in Figure 1D. We replaced the gating process for SCIMP-positive exosomes and control SCIMP-negative exosomes in Figure 1D to more representatively illustrate the differences between nano-flow cytometry and the two types of exosomes.
4. Thanks to the reviewer for patiently pointing out our grammar issues. We have made corrections throughout the entire manuscript.